# Multimodal Molecular Pretraining via Modality Blending

**Qiying Yu**[1,*] **Yudi Zhang**[2*]**, Yuyan Ni**[3]**, Shikun Feng**[1]**, Yanyan Lan**[1,4]**, Hao Zhou**[1,5,†]**, Jingjing Liu**[1]

[1] Institute for AI Industry Research, Tsinghua University     [2] Harbin Institute of Technology
[3] Academy of Mathematics and Systems Science, Chinese Academy of Sciences
[4] Beijing Academy of Artificial Intelligence
[5] Shanghai Artificial Intelligence Laboratory
`yuqy22@mails.tsinghua.edu.cn, zhouhao@air.tsinghua.edu.cn`

## ABSTRACT

Self-supervised learning has recently gained growing interest in molecular modeling for scientific tasks such as AI-assisted drug discovery. Current studies consider leveraging both 2D and 3D molecular structures for representation learning. However, relying on straightforward alignment strategies that treat each modality separately, these methods fail to exploit the intrinsic correlation between 2D and 3D representations that reflect the underlying structural characteristics of molecules, and only perform coarse-grained molecule-level alignment. To derive fine-grained alignment and promote structural molecule understanding, we introduce an atomic-relation level "blend-then-predict" self-supervised learning approach, MOLEBLEND, which first blends atom relations represented by different modalities into one unified relation matrix for joint encoding, then recovers modality-specific information for 2D and 3D structures individually. By treating atom relationships as anchors, MOLEBLEND organically aligns and integrates visually dissimilar 2D and 3D modalities of the same molecule at fine-grained atomic level, painting a more comprehensive depiction of each molecule. Extensive experiments show that MOLEBLEND achieves state-of-the-art performance across major 2D/3D molecular benchmarks. We further provide theoretical insights from the perspective of mutual-information maximization, demonstrating that our method unifies contrastive, generative (cross-modality prediction) and mask-then-predict (single-modality prediction) objectives into one single cohesive framework.

## 1 INTRODUCTION

Self-supervised learning has been successfully applied to molecular representation learning (Xia et al., 2023; Chithrananda et al., 2020), where meaningful representations are extracted from a large amount of unlabeled molecules. The learned representation can then be finetuned to support diverse downstream molecular tasks. Early works design learning objectives based on a single modality (2D topological graphs (Hu et al., 2020; Rong et al., 2020; You et al., 2020), or 3D spatial structures (Zaidi et al., 2022; Liu et al., 2022a; Zhou et al., 2023)). Recently, multimodal molecular pretraining that exploits both 2D and 3D modalities in a single framework (Liu et al., 2022b; Stärk et al., 2022; Liu et al., 2023; Luo et al., 2022; Zhu et al., 2022) has emerged as an alternative solution.

Multimodal pretraining aims to align representations from different modalities. Most existing methods naturally adopt two models (Figure 1(a)) to encode 2D and 3D information separately (Liu et al., 2022b; Stärk et al., 2022; Liu et al., 2023). Contrastive learning is typically employed to *attract* representations of 2D graphs with their corresponding 3D conformations of the same molecule, and *repulse* those from different molecules. Another school of study is generative methods that bridge 2D and 3D modalities via mutual prediction (Figure 1(a-b)), such as taking 2D graphs as input to predict 3D information, and vice versa (Liu et al., 2022b; Zhu et al., 2022; Liu et al., 2023).

---

*Equal contribution. [†] Corresponding Author

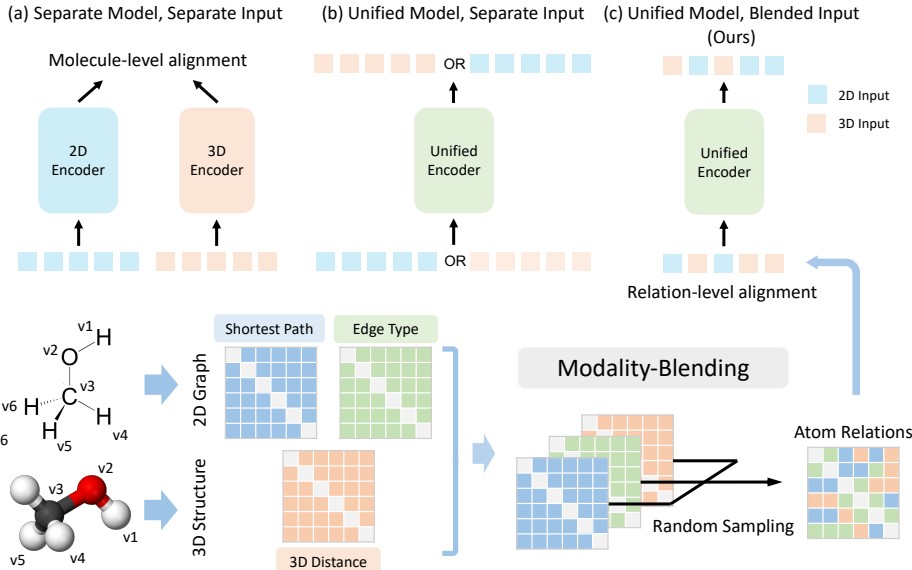

Figure 1: Comparison on the process of input data. (a) (Liu et al., 2021a; Stärk et al., 2022; Liu et al., 2023) and (b) (Zhu et al., 2022) treat different modalities separately, while (c) (ours) blends modalities as input and output. Same atoms ($v1$, ..., $v6$) are shared across modalities, while the depictions of atom relationships (shortest path, edge type, 3D distance) are represented by different matrices, which are blended into an integral input for unified pretraining with explicit alignment.

However, these approaches only align different modalities on a coarse-grained molecule-level. The contrastive learning used in most existing methods has been proved to lack detailed structural understanding of the data (Yuksekgonul et al., 2022; Xie et al., 2022), thus missing a deep comprehension of the constituting atoms and relations, which plays a vital role in representing molecules (Schütt et al., 2017; Liu et al., 2021b). Besides, all methods consider different modalities as independent signals in each model and treat them as separate integral inputs (Figure 1(a-b). This practice divides different modalities apart and ignores the underlying correlation between 2D and 3D modalities, only realizing a rudimentary molecule-level alignment.

To derive a more fine-grained alignment and promote structural molecular understanding, a deeper look into the atom-relation-level sub-structures is asked for. We observe that although appearing visually distinct and residing in different high-dimensional spaces, 2D molecular graphs and 3D spatial structures are intrinsically equivalent as they are essentially different manifestations of the same *atoms* and their *relationships*. The differentiating factor of *relationship* appears as chemical bond or shortest path distance in 2D graph, or 3D euclidean distance in 3D structure. Thus, pivoting around *atom relationship* and explicitly leveraging the alignment between modalities to mutually enhance both 2D and 3D representations can be a more natural and effective alignment strategy.

In this work, we introduce a *relation-level* multimodal pretraining method, MOLEBLEND, which explicitly leverages the alignment of atom relations between 2D and 3D structures and blends input signals from different modalities as one unified data structure to pre-train one single model (Figure 1(c)). Specifically, MOLEBLEND consists of a two-stage *blend-then-predict* training procedure: *modality-blended encoding* and *modality-targeted prediction*. During encoding, we blend different depictions of atom relations from 2D and 3D views into one relation matrix. During prediction, the model recovers missing 2D and 3D information as supervision signals. With such a relation-level blending approach, multimodal molecular information is mingled within a unified model, and fine-grained atom-relation alignment in the multimodal input space leads to a deeper structural understanding of molecular makeup. Extensive experiments demonstrate that MOLEBLEND outperforms existing molecular modeling methods across a broad range of 2D and 3D benchmarks. We further provide theoretical insights from the perspective of mutual-information maximization to validate the proposed pretraining objective.

Our contributions are summarized as follows:

- We propose to align molecule 2D and 3D modalities at atomic-relation level, and introduce MOLEBLEND, a multimodal molecular pretraining method that explicitly utilizes the intrinsic correlations between 2D and 3D representations in pretraining.

- Empirically, extensive evaluation demonstrates that MOLEBLEND achieves state-of-the-art performance over diverse 2D and 3D tasks, verifying the effectiveness of relation-level alignment.

- Theoretically, we provide a decomposition analysis of our objective as an explanatory tool, for better understanding of the proposed blend-then-predict learning objective.

## 2 RELATED WORK

Multimodal molecular pretraining (Liu et al., 2022b; Stärk et al., 2022; Zhu et al., 2022; Luo et al., 2022; Liu et al., 2023) leverages both 2D and 3D information to learn molecular representations. It bears a trade-off between cost and performance, as 3D information is vital for molecular property prediction but 3D models tend to be resource-intensive during deployment. Most existing methods utilize two separate models to encode 2D and 3D information (Liu et al., 2022b; Stärk et al., 2022; Liu et al., 2023). Their pretraining methods mostly use contrastive learning (He et al., 2020), which treats 2D graphs with their corresponding 3D conformations as positive views and information from different molecules as negative views for contrasting. Another pretraining method uses generative models to predict one modality based on the input of another modality Liu et al. (2022b; 2023). Zhu et al. (2022) proposes to encode both 2D and 3D inputs within a single GNN model, but different modalities are still treated as separate inputs. We instead propose to leverage atom relations as the anchor to blend different modalities together as an integral input to a single model.

Masked auto-encoding (Vincent et al., 2008) is a widely applied representation learning method (Devlin et al., 2019; He et al., 2022) that removes a portion of the data and learns to predict the missing content *(mask-then-predict)*. Multimodal masking approaches in other multimodal learning areas (*e.g.*, BEiT-3 (Wang et al., 2022a), UNITER (Chen et al., 2020b)) directly concatenate different modalities into a sequence, then predict the masked tokens, without explicit alignment of modalities in the input space. Different from them, MOLEBLEND blends together the elements of different modalities in the input space with explicit alignment.

## 3 MULTIMODAL MOLECULAR PRETRAINING VIA BLENDING

Molecules are typically represented by either 2D molecular graph or 3D spatial structure. Despite their distinct appearances, they depict a common underlying structure, *i.e.*, atoms and their relationships (*e.g.*, shortest path distance and edge type in 2D molecular graph, and Euclidean distance in 3D structure). Naturally, these representations should be unified organically, instead of treated separately with different models, in order to learn the representation of complex chemical relations underneath. We perform explicit relation-level alignment via blending for unifying modalities.

### 3.1 PROBLEM FORMULATION

A molecule $\mathcal{M}$ can be represented as a set of atoms $\mathcal{V} \in \mathbb{R}^{n \times v}$ along with their relationships $\mathcal{R} \in \mathbb{R}^{n \times n \times r}$, where $n$ is the number of atoms, $v$ and $r$ are dimensions of atom and relation feature, respectively. The nature of $\mathcal{R}$ can vary depending on the context. In the commonly used 2D graph representation of molecules, $\mathcal{R}$ is represented by the chemical bonds $\mathcal{E}$, which are the edges of the 2D molecular graph. In 3D scenarios, $\mathcal{R}$ is defined as the relative Euclidean distance $\mathcal{D}$ between atoms.

To leverage both 2D and 3D representations, we adopt the shortest path distance $\mathcal{R}_{\text{spd}}$ and the edge type encoding $\mathcal{R}_{\text{edge}}$ of molecular graph, as well as Euclidean distance $\mathcal{R}_{\text{distance}}$ in 3D space, as three different appearances of atom relations across 2D/3D modalities. And instead of treating each modality separately with individual models, we blend the three representations into a single matrix $\mathcal{R}_{\text{2D\&3D}}$ by randomly sampling each representation for each vector, following a pre-defined multinomial distribution $S$. Our pre-training objective is to maximize the following likelihood:

$$\max \mathbb{E}_S \, P(\mathcal{R}_{\text{spd}}, \mathcal{R}_{\text{edge}}, \mathcal{R}_{\text{distance}} | \mathcal{R}_{\text{2D\&3D},S}, \mathcal{V}) \tag{1}$$

We employ the Transformer model (Vaswani et al., 2017) to parameterize our objective, capitalizing on its ability to incorporate flexible atom relations in a fine-grained fashion through attention bias (Raffel

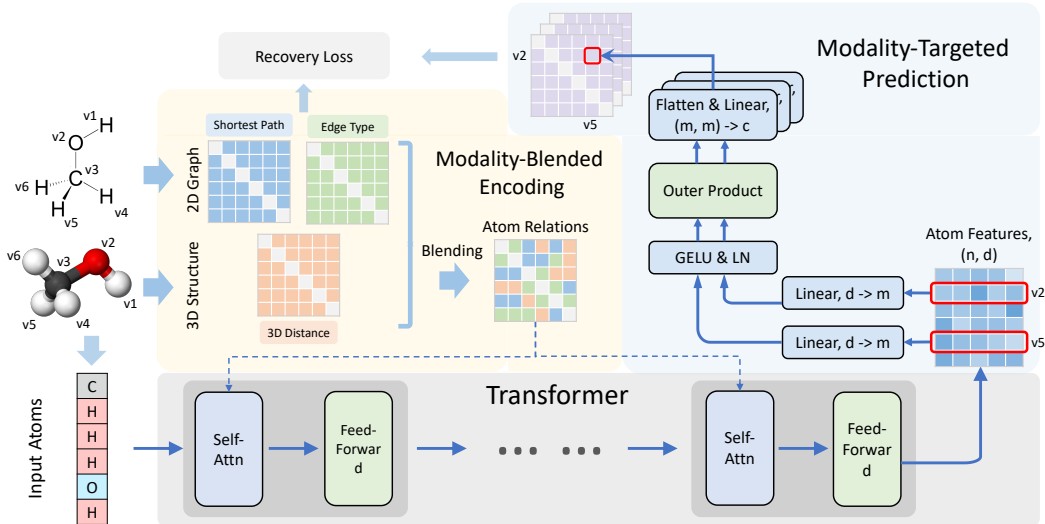

Figure 2: Illustration of unified molecular representation learning process, consisting of two steps: 1) modality-blended encoding, which blends diverse atom relations together and injects it into the self-attention module of Transformer for unified cross-modality encoding; 2) modality-targeted prediction, where atom features encoded by Transformer are transformed into atom relations through an outer product projection module, to recover the diverse relation depictions.

et al., 2020; Shaw et al., 2018; Ke et al., 2021; Ying et al., 2021). This choice is further supported by recent research demonstrating that a single Transformer model can effectively process both 2D and 3D data (Luo et al., 2022).

**Transformer Block**  The Transformer architecture is composed of a stack of identical blocks, each containing a multi-head self-attention layer and a position-wise feed-forward network. Residual connection (He et al., 2016) and layer normalization (Ba et al., 2016) are applied to each layer. Denote $\mathbf{X}^l = [\mathbf{x}_1^l; \mathbf{x}_2^l; \ldots; \mathbf{x}_n^l]$ as the input to the $l$-th block with the sequence length $n$, and each vector $x_i \in \mathbb{R}^d$ is the contextual representation of the atom at position $i$. $d$ is the dimension of the hidden representations. A Transformer block first computes the multi-head self-attention to effectively aggregate the input sequence $\mathbf{X}^l$:

$$\text{Multi-Head}(\mathbf{X}) = \text{Concat}(\text{head}_1, \ldots, \text{head}_h)\mathbf{W}^O \tag{2}$$

where $\text{head}_i = \text{Attention}(\mathbf{X}\mathbf{W}_i^Q, \mathbf{X}\mathbf{W}_i^K, \mathbf{X}\mathbf{W}_i^V)$ and $h$ is the number of attention heads. $\mathbf{W}_i^Q$, $\mathbf{W}_i^K, \mathbf{W}_i^V \in \mathbb{R}^{d \times d_h}, \mathbf{W}^O \in \mathbb{R}^{d \times d}$ are learnable parameter matrices. The attention computation is defined as:

$$\text{Attention}(\mathbf{Q}, \mathbf{K}, \mathbf{V}) = \text{softmax}\left(\frac{\mathbf{Q}\mathbf{K}^\top}{\sqrt{d}}\right)\mathbf{V} \tag{3}$$

Generally, given input $X^l$, the $l$-th block works as follows:

$$\tilde{\mathbf{X}}^l = \text{LayerNorm}\left(\mathbf{X}^l + \text{Multi-Head}(\mathbf{X}^1)\right) \tag{4}$$

$$\mathbf{X}^{l+1} = \text{LayerNorm}\left(\tilde{\mathbf{X}}^l + \text{GELU}(\tilde{\mathbf{X}}^l\mathbf{W}_1^l)\mathbf{W}_2^l\right) \tag{5}$$

where $\mathbf{W}_1^l \in \mathbb{R}^{d \times d_f}, \mathbf{W}_2^l \in \mathbb{R}^{d_f \times d}$, and $d_f$ is the intermediate size of the feed-forward layer.

## 3.2  LEARNING OBJECTIVE

To facilitate fine-grained alignment and organic integration of different depictions of atoms and their relations across 2D/3D spaces, we design a new 'blend-then-predict' training paradigm that consists of two steps: 1) modality-blended encoding that encodes a molecule with blended information from different modalities; and 2) modality-targeted prediction that recovers the original 2D and 3D input.

The pre-training process is illustrated in Figure 2. The core idea is to bind different modalities together at a granular level by blending relations from multiple modalities into an integral input from the get-go, to encourage the model to discover fundamental and unified relation representations across heterogeneous forms.

**Modality-blended Encoding** Multimodal learning aims to learn the most essential representations of data that possess inherent connections while appearing distinctive between different modalities. In the context of molecules, atom relationships are the common attributes underpinning different representations across 2D/3D modalities. This motivates us to leverage relations as anchors, to align both modalities in a fine-grained manner that blends multimodalities from the very beginning.

We adopt three appearances of relations across 2D and 3D modalities following (Luo et al., 2022): shortest path distance, edge type, and 3D Euclidean distance. For each atom pair $(i, j)$, $\Psi_{\text{SPD}}^{ij}$ represents the shortest path distance between atom $i$ and $j$. We encode the edge features along the shortest path between $i$ and $j$ as the edge encoding, $\Psi_{\text{Edge}}^{ij} = \frac{1}{N} \sum_{n=1}^{N} \mathbf{w}_n^\top \mathbf{e}_n$, where $(\mathbf{e}_1, \mathbf{e}_2, \ldots, \mathbf{e}_N), e_n \in \mathbb{R}^{d_e}$ are features of edges on the shortest path between $i$ and $j$. $w_n \in \mathbb{R}^{d_e}$ are learnable parameters. Following (Zhou et al., 2023; Luo et al., 2022), we encode Euclidean distances of an atom pair $(i, j)$ with Gaussian Basis Kernel function (Schölkopf et al., 1997):

$$\zeta_k^{ij} = \mathcal{G}(\mathcal{A}(d^{ij}; \gamma^{ij}, \beta^{ij}); \mu_k, \sigma_k), \ k = 1, \ldots, K \tag{6}$$

$$\Psi_{\text{Distance}}^{ij} = \text{GELU}(\zeta^{ij} \cdot W_{\text{3D}}^1) W_{\text{3D}}^2, \ \zeta^{ij} = [\zeta_1^{ij}, \ldots, \zeta_K^{ij}]^\top \tag{7}$$

where $\mathcal{A}(d; \gamma, \beta) = \gamma \cdot d + \beta$ is the affine transformation with learnable parameters $\gamma$ and $\beta$, and $\mathcal{G}(d; \mu, \sigma) = \frac{1}{\sqrt{2\pi}\sigma} \exp\left(-\frac{1}{2\sigma^2}(d - \mu)^2\right)$ is the Gaussian density function with parameters $\mu$ and $\sigma$. $K$ is the number of Gaussian Basis kernels. $W_{\text{3D}}^1 \in \mathbb{R}^{K \times K}, W_{\text{3D}}^2 \in \mathbb{R}^{K \times 1}$ are learnable parameters. $\Psi_{\text{SPD}}, \Psi_{\text{Edge}}, \Psi_{\text{Distance}}$ denote the three relation matrices of all atom pairs, with the same shape $n \times n$.

Different from existing works that separately feed one of these relations into different models, we blend them together from the get-go and randomly mix them into one relation matrix, which is then fed into one single model for molecule encoding. Specifically, we first define a multinomial distribution $S$ with a probability vector $p = (p_1, p_2, p_3)$. For each position $(i, j)$ in the matrix, we draw a sample $s^{ij} \in \{1, 2, 3\}$ following the probability distribution $p$, then determine the corresponding element of the blended matrix as follows:

$$\Psi_{\text{2D\&3D}}^{ij} = \Psi_{\text{SPD}}^{ij} \mathbb{1}_1 + \Psi_{\text{Edge}}^{ij} \mathbb{1}_2 + \Psi_{\text{Distance}}^{ij} \mathbb{1}_3, \ \text{where} \ \mathbb{1}_k = \begin{cases} 1 & \text{if } s^{ij} = k \\ 0 & \text{otherwise} \end{cases} \tag{8}$$

, where each position $(i, j)$ randomly selects its element from one of the $\Psi_{\text{SPD}}^{ij}, \Psi_{\text{Edge}}^{ij}, \Psi_{\text{Distance}}^{ij}$. After the process finishes, distinct relation manifestations ($\Psi_{\text{SPD}}, \Psi_{\text{Edge}}, \Psi_{\text{Distance}}$) across modalities are blended into a single modality-blended matrix $\Psi_{\text{2D\&3D}} \in \mathbb{R}^{n \times n}$ without overlapping sub-structures, to represent the inter-atomic relations.

We inject this *modality-blended* relation $\Psi_{\text{2D\&3D}}$ into the self-attention module, which captures pair-wise relations between inputs atoms, to provide complementary pair-wise information. This practice is also similar to the relative positional encoding for Transformer (Raffel et al., 2020):

$$\text{Attention}(\mathbf{Q}, \mathbf{K}, \mathbf{V}) = \text{softmax}\left(\frac{\mathbf{Q}\mathbf{K}^\top}{\sqrt{d}} + \Psi_{\text{2D\&3D}}\right) \mathbf{V} \tag{9}$$

With modality-blending, we explicitly bind different modalities together at fine-grained relation level, which will help the model better integrate and align modalities at fine-grained level.

**Modality-targeted Prediction** The model recovers the full $\mathcal{R}_{\text{spd}}, \mathcal{R}_{\text{edge}}$ and $\mathcal{R}_{\text{distance}}$ as its training objectives. The intuition is, if the model can predict different types of atom relations, like shortest path on the molecular graph or 3D Euclidean distance, given *a single mixed representation*, this cross-modality representation must have captured some underlying integral molecular structure.

Specifically, after modality-blended encoding, we obtain contextual atom representations $\mathbf{X}^{L+1} \in \mathbb{R}^{n \times d}$ encoded by an $L$-layer Transformer. We propose an outer product projection module to transform the atom representations into $n \times n$ atom relations. The representations $\mathbf{X}^{L+1}$ are first

linearly projected to a smaller dimension $m = 32$ with two independent Linear layers $\mathbf{W}_l, \mathbf{W}_r \in \mathbb{R}^{m \times d}$. The outer products are computed upon the transformed representations, which are then flattened and projected into the target space with a modality-targeted head $\mathbf{W}_{\text{head}} \in \mathbb{R}^{c \times m^2}$. The relation computation between the $i$-th and $j$-th atoms is formulated as follows:

$$\mathbf{o}_{ij} = \mathrm{G}(\mathbf{W}_l \mathbf{X}_i^{L+1}) \otimes \mathrm{G}(\mathbf{W}_r \mathbf{X}_j^{L+1})^\top \in \mathbb{R}^{m \times m} \tag{10}$$

$$\mathbf{z}_{ij} = \mathbf{W}_{\text{head}} \cdot \text{Flatten}(\mathbf{o}_{ij}) \in \mathbb{R}^c \tag{11}$$

where $\mathrm{G}(\cdot) = \text{LayerNorm}(\text{GELU}(\cdot))$. We now obtain the modality-targeted relation matrix $\mathbf{Z} \in \mathbb{R}^{n \times n \times c}$, where $c$ depends on the targeted task. The predictions of shortest path distance and edge type are formulated as classification tasks, where $c$ is the number of possible shortest path distance or edge types. For predicting 3D distance, we formulate it as a 3-dimensional regression task, and the regression targets are the relative Euclidean distances in 3D space.

**Noisy Node as Regularization**  Noisy node (Godwin et al., 2022; Zaidi et al., 2022; Luo et al., 2022) incorporates an auxiliary loss for coordinate denoising in addition to the original objective, which has been found effective in improving representation learning. We also adopt this practice as an additional regularization term, by adding Gaussian noise to the input coordinates and requiring the model to predict the added noise.

### 3.3  FINETUNING

The trained model can be finetuned to accept both 2D and 3D inputs for downstream tasks. For scenarios where a large amount of 2D molecular graphs is available while 3D conformations are too expensive to obtain, the model can take only 2D input to finetune the model. Formally, given shortest path distance $\mathcal{R}_{\text{spd}}$, edge type $\mathcal{R}_{\text{edge}}$ and atom types $\mathcal{V}$ as available 2D information, we define $\mathbf{y}_{\text{2D}}$ as the task target, $K$ as the number of training samples, and $\ell(\cdot, \cdot)$ as the loss function of the specific training task. The 2D finetuning objective is then defined as:

$$\mathcal{L}_{\text{2D}} = \frac{1}{K} \sum_{k=1}^{K} \ell \left( f(\mathcal{R}_{\text{spd}}^k, \mathcal{R}_{\text{edge}}^k, \mathcal{V}^k), \mathbf{y}_{\text{2D}}^k \right) \tag{12}$$

When it comes to scenarios where 3D information is obtained, we propose to incorporate both 2D and 3D information as model input, as generating 2D molecular graphs from 3D conformations is free and can bring in useful information from 2D perspective. The multimodal input is injected into the self-attention module that captures pair-wise relations:

$$\text{Attention}(\mathbf{Q}, \mathbf{K}, \mathbf{V}) = \text{softmax}\left( \frac{\mathbf{Q}\mathbf{K}^\top}{\sqrt{d}} + \Psi_{\text{SPD}} + \Psi_{\text{Edge}} + \Psi_{\text{Distance}} \right) \mathbf{V} \tag{13}$$

$$\mathcal{L}_{\text{3D}} = \frac{1}{K} \sum_{k=1}^{K} \ell \left( f(\mathcal{R}_{\text{spd}}^k, \mathcal{R}_{\text{edge}}^k, \mathcal{R}_{\text{distance}}^k, \mathcal{V}^k), \mathbf{y}_{\text{3D}}^k \right) \tag{14}$$

This practice is unique in utilizing information from multiple modalities for a single-modality task, which is infeasible in previous 3D (Zaidi et al., 2022) or multimodal methods with separate models for different modalities (Liu et al., 2022b; Stärk et al., 2022; Liu et al., 2023). Empirically, we find that the integration of 2D information helps improve performance. we hypothesize that: 1) 2D information, such as chemical bond on a molecular graph, encodes domain experts' prior knowledge and provides references to 3D structure; 2) 3D structures obtained from computational simulations can suffer from inevitable approximation errors (Luo et al., 2022) which are avoided in our approach.

### 3.4  THEORETICAL INSIGHTS

In this section, we present a theoretical perspective from mutual information (MI) maximization for a better understanding of the 'blend-then-predict' process. We demonstrate that this approach unifies existing contrastive, generative (inter-modality prediction), and mask-then-predict (intra-modality prediction) objectives within a single objective formulation.

For simplicity, we consider two relations, denoted as $\mathcal{R}_{\text{2D}} = (a_{ij})_{n \times n}$ and $\mathcal{R}_{\text{3D}} = (b_{ij})_{n \times n}$. Their elements are randomly partitioned into two parts, represented as $\mathcal{R}_{\text{2D}} = [A_1, A_2], \mathcal{R}_{\text{3D}} = [B_1, B_2]$,

such that $A_i$ shares identical elements indexes with $B_i$, $i \in \{1,2\}$. The blended matrix is denoted as $\mathcal{R}_{\text{2D\&3D}} = [A_1, B_2]$.

**Proposition 3.1 (Mutual information Maximization)** *The training process with modality-blending maximizes the lower bound of the following mutual information: $\mathbb{E}_S I(A_2; A_1, B_2) + I(B_1; A_1, B_2)$. The proof can be found in Appendix B.2.4.*

**Proposition 3.2 (Mutual Information Decomposition)** *The mutual information $I(A_2; A_1, B_2) + I(B_1; A_1, B_2)$ can be decomposed into two components below. The first one corresponds to the objectives of **contrastive and generative** approaches. The second component, the primary focus of our research, represents the **mask-then-predict** objective (proof in Proposition B.1 in Appendix):*

$$
\begin{aligned}
I(A_2; A_1, B_2) + I(B_1; A_1, B_2) = &\frac{1}{2}[\underbrace{I(A_1; B_1) + I(A_2; B_2)}_{\text{contrastive and generative}} + \underbrace{I(A_1; B_1|B_2) + I(A_2; B_2|A_1)}_{\text{conditional contrastive and generative}}] \\
&+ \frac{1}{2}[\underbrace{I(A_1; A_2) + I(B_1; B_2)}_{\text{mask-then-predict}} + \underbrace{I(A_1; A_2|B_2) + I(B_1; B_2|A_1)}_{\text{multimodal mask-then-predict}}]
\end{aligned}
\tag{15}
$$

The first part of Equation 15 corresponds to existing (conditional) contrastive and generative methods, which aim to maximize the MI between two corresponding parts ($A_i$ with $B_i$, $i \in \{1,2\}$) across two modalities (see Appendix B.2.1 and B.2.3 for the detailed proof). The second part represents the (multimodal) mask-then-predict objectives, focusing on maximizing the mutual information between the masked and the remaining parts within a single modality (refer to Appendix B.2.2 for details).

This decomposition illustrates that our objective unifies contrastive, generative (inter-modality prediction), and mask-then-predict (intra-modality prediction) approaches within a single cohesive *blend-then-predict* framework, from the perspective of MI maximization. Moreover, this approach fosters enhanced cross-modal interaction with an innovative multimodal mask-then-predict target.

## 4 EXPERIMENTS

### 4.1 EXPERIMENTAL SETUP

**Datasets.** For pretraining, we use PCQM4Mv2 dataset from the OGB Large-Scale Challenge (Hu et al., 2021), which includes 3.37 million molecules with both 2D graphs and 3D geometric structures. To evaluate the versatility of MOLEBLEND, we carry out extensive experiments on 24 molecular tasks with different data formats across three representative benchmarks: MoleculeNet (Wu et al., 2017) (2D, 11 tasks), QM9 quantum properties (Ramakrishnan et al., 2014) (3D, 12 tasks), and PCQM4Mv2 humo-lumo gap (2D). Further details about these datasets can be found in the Appendix C.1.

**Baselines.** We choose the most representative 2D and 3D pretraining baselines: AttrMask (Hu et al., 2020), ContextPred (Hu et al., 2020), InfoGraph (Sun et al., 2020), MolCLR (Wang et al., 2022b), GraphCL (You et al., 2020), GraphLoG (Xu et al., 2021), MGSSL Zhang et al. (2021), as well as recently published method Mole-BERT (Xia et al., 2023) and GraphMAE (Hou et al., 2022) as 2D baselines. In addition, we adopt GraphMVP (Liu et al., 2022b), 3D InfoMax (Stärk et al., 2022), UnifiedMol Zhu et al. (2022) and MoleculeSDE (Liu et al., 2023) as multimodal baselines. As most baselines adopt GNN as backbone, we further implement two close-related multimodal pretraining baselines, 3D Infomax and GraphMVP, under the same Transformer backbone as we use, to fairly compare the effectiveness of pretraining objective.

**Backbone Model.** Following (Ying et al., 2021; Luo et al., 2022), we employ a 12-layer Transformer of hidden size 768, with 32 attention heads. For pretraining, we use AdamW optimizer and set $(\beta_1, \beta_2)$ to (0.9, 0.999) and peak learning rate to 1e-5. Batch size is 4096. We pretrain the model for 1 million steps with initial 100k steps as warm-up, after which learning rate decreases to zero with cosine scheduler. The blending ratio $p$ is 2:2:6, and the ablations on $p$ can be found in Appedix A.3.

### 4.2 EVALUATION ON 2D CAPABILITY

We evaluate MOLEBLEND on MoleculeNet, one of the most widely used benchmarks for 2D molecular property prediction, which covers molecular properties ranging from quantum mechanics

Table 1: Results on molecular property classification tasks (with 2D topology only). We report ROC-AUC score (higher is better) under scaffold splitting. Transformer impl. represents implementation under the same Transformer backbone as MOLEBLEND. Results in gray are evaluated under a different protocol.

| Pre-training Methods | Backbone Type | BBBP ↑ | Tox21 ↑ | ToxCast ↑ | SIDER ↑ | ClinTox ↑ | MUV ↑ | HIV ↑ | Bace ↑ | Avg ↑ |
|---|---|---|---|---|---|---|---|---|---|---|
| AttrMask (Hu et al., 2020) | GNN | 65.0±2.3 | 74.8±0.2 | 62.9±0.1 | 61.2±0.1 | 87.7±1.1 | 73.4±2.0 | 76.8±0.5 | 79.7±0.3 | 72.68 |
| ContextPred (Hu et al., 2020) | GNN | 65.7±0.6 | 74.2±0.0 | 62.5±0.3 | 62.2±0.5 | 77.2±0.8 | 75.3±1.5 | 77.1±0.8 | 76.0±2.0 | 71.28 |
| GraphCL (You et al., 2020) | GNN | 69.7±0.6 | 73.9±0.6 | 62.4±0.5 | 60.5±0.8 | 76.0±2.6 | 69.8±2.6 | 78.5±1.2 | 75.4±1.4 | 70.78 |
| InfoGraph (Sun et al., 2020) | GNN | 67.5±0.1 | 73.2±0.4 | 63.7±0.5 | 59.9±0.3 | 76.5±1.0 | 74.1±0.7 | 75.1±0.9 | 77.8±0.8 | 70.98 |
| GROVER (Rong et al., 2020) | Transformer | 70.0±0.10 | 74.3±0.1 | 65.4±0.4 | 64.8±0.6 | 81.2±3.0 | 67.3±1.8 | 62.5±0.9 | 82.6±0.7 | 71.01 |
| MolCLR (Wang et al., 2022b) | GNN | 66.6±1.8 | 73.0±0.1 | 62.9±0.3 | 57.5±1.7 | 86.1±0.9 | 72.5±2.3 | 76.2±1.5 | 71.5±3.1 | 70.79 |
| GraphLoG (Xu et al., 2021) | GNN | 72.5±0.8 | 75.7±0.5 | 63.5±0.7 | 61.2±1.1 | 76.7±3.3 | 76.0±1.1 | 77.8±0.8 | 83.5±1.2 | 73.40 |
| MGSSL (Zhang et al., 2021) | GNN | 69.7±0.9 | 76.5±0.3 | 64.1±0.7 | 61.8±0.8 | 80.7±2.1 | 78.7±1.5 | 78.8±1.2 | 79.1±0.9 | 73.70 |
| GraphMAE (Hou et al., 2022) | GNN | 72.0±0.6 | 75.5±0.6 | 64.1±0.3 | 60.3±1.1 | 82.3±1.2 | 76.3±2.4 | 77.2±1.0 | 83.1±0.9 | 73.85 |
| Mole-BERT (Xia et al., 2023) | GNN | 71.9±1.6 | 76.8±0.5 | 64.3±0.2 | 62.8±1.1 | 78.9±3.0 | 78.6±1.8 | 78.2±0.8 | 80.8±1.4 | 74.04 |
| 3D InfoMax (Stärk et al., 2022) | GNN | 69.1±1.0 | 74.5±0.7 | 64.4±0.8 | 60.6±0.7 | 79.9±3.4 | 74.4±2.4 | 76.1±1.3 | 79.7±1.5 | 72.34 |
| GraphMVP (Liu et al., 2022b) | GNN | 68.5±0.2 | 74.5±0.4 | 62.7±0.1 | 62.3±1.6 | 79.0±2.5 | 75.0±1.4 | 74.8±1.4 | 76.8±1.1 | 71.69 |
| MoleculeSDE (Liu et al., 2023) | GNN | 71.8±0.7 | 76.8±0.3 | 65.0±0.2 | 60.8±0.3 | 87.0±0.5 | **80.9±0.3** | 78.8±0.9 | 79.5±2.1 | 75.07 |
| Transformer from scratch | Transformer | 69.4±1.1 | 74.2±0.3 | 62.6±0.3 | **65.8±0.3** | **90.3±0.9** | 71.3±0.8 | 76.2±0.6 | 79.5±0.2 | 73.66 |
| 3D InfoMax (Transformer impl.) | Transformer | 70.4±1.0 | 75.5±0.5 | 63.1±0.7 | 64.1±0.1 | 89.8±1.2 | 72.8±1.0 | 74.9±0.3 | 80.7±0.6 | 73.91 |
| GraphMVP (Transformer impl.) | Transformer | 71.5±1.3 | 76.1±0.9 | 64.3±0.6 | 64.7±0.7 | 89.9±0.9 | 74.9±1.2 | 76.0±0.6 | 81.5±1.2 | 74.86 |
| MOLEBLEND | Transformer | **73.0±0.8** | **77.8±0.8** | **66.1±0.0** | 64.9±0.3 | 87.6±0.7 | 77.2±2.3 | **79.0±0.8** | **83.7±1.4** | **76.16** |

and physical chemistry to biophysics and physiology. We use the scaffold split (Wu et al., 2017), and report the mean and standard deviation of results of 3 random seeds.

Table 1 presents the ROC-AUC scores for all compared methods on eight classification tasks. Remarkably, MOLEBLEND achieves state-of-the-art performance in 5 out of 8 tasks, with significant margins in some cases (*e.g.*, 83.7 v.s. 81.5 on Bace). Note that all other multimodal methods (3D Infomax (Stärk et al., 2022), GraphMVP (Liu et al., 2022b), MoleculeSDE (Liu et al., 2023)) utilize two separate modality-specific models, with contrastive learning as one of their objectives. In contrast, MOLEBLEND models molecules in a *unified* manner, and perform 2D and 3D alignment in a *fine-grained* relation-level, demonstrating superior performance. MOLEBLEND also outperforms all 2D baselines (upper section of the table), demonstrating that incorporating 3D information helps improve the prediction of molecular properties. Table 6 summarizes the performance of different methods on three regression tasks of MoleculeNet, which substantiates the superiority of MOLEBLEND.

### 4.3 EVALUATION ON 3D CAPABILITY

We use QM9 (Ramakrishnan et al., 2014) dataset to evaluate the effectiveness of MOLEBLEND on 3D tasks. QM9 is a quantum chemistry benchmark with 134K small organic molecules. It contains 12 tasks, covering the energetic, electronic and thermodynamic properties of molecules. Following (Thölke & Fabritiis, 2022), we randomly split 10,000 and 10,831 molecules as validation and test set, and use the remaining molecules for finetuning. Results are presented in Table 2, evaluated on MAE metric (lower is better). MOLEBLEND achieves state-of-the-art performance

Table 2: Results on QM9 datasets. Mean Absolute Error (MAE, lower is better) is reported.

| Pre-training Methods | Alpha ↓ | Gap ↓ | HOMO ↓ | LUMO ↓ | Mu ↓ | Cv ↓ | G298 ↓ | H298 ↓ | R2 ↓ | U298 ↓ | U0 ↓ | Zpve ↓ |
|---|---|---|---|---|---|---|---|---|---|---|---|---|
| Distance Prediction (Liu et al., 2022a) | 0.065 | 45.87 | 27.61 | 23.34 | 0.031 | 0.033 | 14.83 | 15.81 | 0.248 | 15.07 | 15.01 | 1.837 |
| 3D InfoGraph (Liu et al., 2022a) | 0.062 | 45.96 | 29.29 | 24.60 | 0.028 | 0.030 | 13.93 | 13.97 | **0.133** | 13.55 | 13.47 | 1.644 |
| 3D InfoMax (Stärk et al., 2022) | 0.057 | 42.09 | 25.90 | 21.60 | 0.028 | 0.030 | 13.73 | 13.62 | 0.141 | 13.81 | 13.30 | 1.670 |
| GraphMVP (Liu et al., 2022b) | 0.056 | 41.99 | 25.75 | 21.58 | 0.027 | 0.029 | 13.43 | 13.31 | 0.136 | 13.03 | 13.07 | 1.609 |
| MoleculeSDE (Liu et al., 2023) | **0.054** | 41.77 | 25.74 | 21.41 | **0.026** | **0.028** | 13.07 | 12.05 | 0.151 | 12.54 | 12.04 | 1.587 |
| MOLEBLEND | 0.060 | **34.75** | **21.47** | **19.23** | 0.037 | 0.031 | **12.44** | **11.97** | 0.417 | **12.02** | **11.82** | **1.580** |

Table 3: Ablation studies on pretraining objectives. The best and second best results are marked by **bold** and underlined.

| Pre-training Methods | BBBP ↑ | Tox21 ↑ | ToxCast ↑ | SIDER ↑ | ClinTox ↑ | MUV ↑ | HIV ↑ | Bace ↑ | U298 ↓ | U0 ↓ |
|---|---|---|---|---|---|---|---|---|---|---|
| Noisy-Node | 68.50 | 76.25 | 65.48 | 63.71 | 83.28 | **78.80** | 79.13 | 82.72 | 14.31 | 13.80 |
| Blend-then-Predict | 71.59 | 75.61 | 65.93 | 64.58 | **90.82** | 76.81 | **79.74** | 83.53 | 14.56 | 15.35 |
| MOLEBLEND | **73.00** | **77.82** | **66.14** | **64.90** | 87.62 | 77.23 | 79.01 | **83.66** | **12.02** | **11.82** |

among multimodal methods on 8 out of 12 tasks, some of which with a large margin (*e.g.*, Gap, HOMO, LUMO), demonstrating the strong capability of our model for 3D tasks.

## 4.4 ABLATION STUDIES

**Pretraining Objectives**  Table 3 studies the effect of different pretraining objectives: *noisy-node*, *blend-then-predict*, and blend-then-predict with noisy-node as regularization (MOLEBLEND). We observe that in most tasks, combining *blend-then-predict* and *noisy-node* yields better representations. In 2D scenarios, we find that *blend-then-predict* outperforms *noisy-node* on 5 out of 8 tasks studied, demonstrating its strong ability to process 2D inputs. While on 3D tasks (U298 and U0), *blend-then-predict* typically performs worse than *noisy-node*. This is because *noisy-node* is a pure 3D denoising task, which makes it more suitable for 3D tasks.

Table 4: Ablation studies on blending vs masking.

| Method | BBBP ↑ | BACE ↑ | Tox21 ↑ | ToxCast ↑ |
|---|---|---|---|---|
| SPD mask | 68.95 | 80.64 | 75.59 | 62.82 |
| Edge mask | 69.02 | 81.97 | 76.01 | 63.81 |
| 3D mask | 67.60 | 80.35 | 75.65 | 63.28 |
| Blending | **71.68** | **83.41** | **76.58** | **65.46** |

**Blending vs Single-modality Mask-then-Predict**  Table 4 studies the effect of multimodal blending compared to single-modality mask-then-predict (SPD, Edge, and 3D mask). We trained all models for 200K steps, keeping all settings consistent except for the learning objective. The results demonstrate that modality blending achieves better performance over modality-specific mask-then-predict.

**Finetuning Settings**  When 3D molecular information is provided, we propose to incorporate both 2D topological and 3D structural information into the model, as generating 2D molecular graphs from 3D conformations is computationally inexpensive. Table 5 demonstrates that the inclusion of 2D information leads to a noticeable improvement in performance. We hypothesize that this is due to the fact that 2D information encodes chemical bond and connectivity on a molecular

Table 5: Ablation studies on fintuning settings of 3D tasks.

| Finetune Settings | Alpha ↓ | HOMO ↓ | Mu ↓ |
|---|---|---|---|
| 3D | 0.066 | 23.62 | 0.042 |
| 3D + 2D | **0.060** | **21.47** | **0.037** |

graph, which is grounded in prior knowledge of domain experts and contains valuable references to 3D structure. Note that this practice is a unique advantage of MOLEBLEND, as we pretrain with both 2D and 3D information blended as one single input into a unified model, which is not feasible in previous multimodal methods that utilize two distinct models for 2D and 3D modalities.

## 5 CONCLUSION

We propose MOLEBLEND, a novel relation-level self-supervised learning method for unified molecular modeling that organically integrates 2D and 3D modalities in a fine-grained manner. By treating atom relations as the anchor, we blend different modalities into an integral input for pretraining, which overcomes the limitations of existing approaches that distinguish 2D and 3D modalities as independent signals. Extensive experimental results reveal that MOLEBLEND achieves state-of-the-art performance on a wide range of 2D and 3D benchmarks, demonstrating the superiority of fine-grained alignment of different modalities.

ACKNOWLEDGEMENT

This work is supported by the National Key R&D Program of China (2022ZD0160501), Natural Science Foundation of China (62376133) and Beijing Academy of Artificial Intelligence (BAAI).

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

# A    EXPERIMENTS

## A.1    BASELINE RESULTS

The baseline results of GraphMVP Liu et al. (2021a), MoleSDE Liu et al. (2023), GraphCL You et al. (2020), GraphMAE Hou et al. (2022), GraphLoG (Xu et al., 2021), MGSSL (Zhang et al., 2021) are from their own paper. Results of AttrMask (Hu et al., 2020), ContextPred (Hu et al., 2020), InfoGraph Sun et al. (2020), MolCLR Wang et al. (2022b) are from MoleculeSDE Liu et al. (2023). Results of MoleBERT (Xia et al., 2023), 3D Infomax Stärk et al. (2022) are from MoleBERT. The results of GROVER Rong et al. (2020) are from Uni-Mol Zhou et al. (2023).

## A.2    MOLNET REGRESSION TASK

Table 6 presents the performance of different methods on three regression tasks of MoleculeNet. In all these tasks, MOLEBLEND achieves state-of-the-art performance, further substantiating the superiority of unified fine-grained molecular modeling.

Table 6: Results on molecular property prediction regression tasks (with 2D topology only). We report RMSE (lower is better) for each task.

| Pre-training Methods | ESOL $\downarrow$ | FreeSolv $\downarrow$ | Lipo $\downarrow$ |
|---|---|---|---|
| AttrMask (Hu et al., 2020) | 1.112±0.048 | - | 0.730±0.004 |
| ContextPred (Hu et al., 2020) | 1.196±0.037 | - | 0.702±0.020 |
| GROVER$_{base}$ (Rong et al., 2020) | 0.983±0.090 | 2.176±0.052 | 0.817±0.008 |
| MolCLR (Wang et al., 2022b) | 1.271±0.040 | 2.594±0.249 | 0.691±0.004 |
| 3D InfoMax (Stärk et al., 2022) | 0.894±0.028 | 2.337±0.227 | 0.695±0.012 |
| GraphMVP (Liu et al., 2022b) | 1.029±0.033 | - | 0.681±0.010 |
| MOLEBLEND | **0.831±0.026** | **1.910±0.163** | **0.638±0.004** |

## A.3    ABLATIONS ON BLENDING RATIO

Table 7 presents ablations on the relation blending ratio, showing that model performance is robust to the random ratio of multinomial distribution. In these experiments, we trained all models for 200K steps, maintaining other settings unchanged (e.g., learning rate consistent), with the exception of the blending ratio.

Furthermore, we have observed that a higher 3D distance ratio (referring to the bottom three rows in the table) sometimes performs better than lower ratio (top row of 4:4:2 ratio). This suggests that the inclusion of 3D information is potentially more important for enhancing the model's understanding of molecular properties. However, it is worth noting that the disparity in performance between these ratios is relatively minor.

Table 7: Ablations on the blending ratio.

| SPD:Edge:3D ($p$) | BBBP $\uparrow$ | BACE $\uparrow$ | Tox21 $\uparrow$ | ToxCast $\uparrow$ | Lipo $\downarrow$ |
|---|---|---|---|---|---|
| 4:4:2 | 72.25 | 82.17 | 76.23 | **66.70** | 0.7544 |
| 3:3:4 | 72.34 | 82.47 | **77.19** | 66.16 | 0.7505 |
| 2:2:6 | **72.52** | **82.89** | 76.15 | 66.58 | 0.7511 |
| 1:1:8 | 72.45 | 82.43 | 76.46 | 66.57 | **0.7478** |

# B THEORETICAL ANALYSIS

In the following sections, we follow common notations(Cover & Thomas, 1991), using uppercase letters to represent random variables and lowercase letters to represent samples of the random variables.

## B.1 MISSING PROOFS

**Lemma B.1 (Chain rule of mutual information(Cover & Thomas, 1991))**

$$I(X_1, X_2; Y) = I(X_1; Y) + I(X_2; Y|X_1) \tag{16}$$

**Proof**

$$
\begin{aligned}
I(X_1; Y) + I(X_2; Y|X_1) &= E_{p(x_1,y)}\big[\log \frac{p(x_1,y)}{p(x_1)p(y)}\big] + E_{p(x_1,x_2,y)}\big[\log \frac{p(x_2,y|x_1)}{p(x_2|x_1)p(y|x_1)}\big] \\
&= E_{p(x_1,x_2,y)}\big[\log \frac{p(x_1,y)}{p(x_1)p(y)}\frac{p(x_2,y|x_1)}{p(x_2|x_1)p(y|x_1)}\big] \\
&= E_{p(x_1,x_2,y)}\big[\log \frac{p(x_1,y)p(x_2,y,x_1)}{p(y)p(x_2,x_1)p(y,x_1)}\big] \\
&= E_{p(x_1,x_2,y)}\big[\log \frac{p(x_2,y,x_1)}{p(y)p(x_2,x_1)}\big] = I(X_1, X_2; Y)
\end{aligned}
\tag{17}
$$

**Proposition B.1 (Mutual Information Decomposition)** *The blend-and-predict method is maximizing the lower bound of the mutual information target below, which can be further divided into two parts.*

$$
\begin{aligned}
&I(A_2; A_1, B_2) + I(B_1; A_1, B_2) \\
=&\frac{1}{2}\big[I(A_1; B_1) + I(A_2; B_2) + I(A_1; B_1|B_2) + I(A_2; B_2|A_1)\big] + \\
&\frac{1}{2}\big[I(A_1; A_2) + I(B_1; B_2) + I(A_1; A_2|B_2) + I(B_1; B_2|A_1)\big]
\end{aligned}
\tag{18}
$$

**Proof** *Firstly, we provide the decomposition of first term in equation 18, i.e. $I(A_2; A_1, B_2)$. By using Lemma B.1 and letting $X_1 = A_1$, $X_2 = B_2$ and $Y = A_2$, we have*

$$I(A_2; A_1, B_2) = I(A_1; A_2) + I(A_2; B_2|A_1). \tag{19}$$

*Again use Lemma B.1 and let $X_1 = B_2$, $X_2 = A_1$ and $Y = A_2$, then we have*

$$I(A_2; A_1, B_2) = I(B_2; A_2) + I(A_2; A_1|B_2). \tag{20}$$

*From equation 19 and equation 20, we have*

$$I(A_2; A_1, B_2) = \frac{1}{2}\big[I(A_1; A_2) + I(A_2; B_2|A_1) + I(B_2; A_2) + I(A_2; A_1|B_2)\big]. \tag{21}$$

*Similarly, we apply Lemma B.1 to decompose the second term in equation 18.*

$$I(B_1; A_1, B_2) = \frac{1}{2}\big[I(B_1; A_1) + I(B_2; B_2|A_1) + I(B_1; B_2) + I(B_1; A_1|B_2)\big]. \tag{22}$$

*End of proof.*

## B.2 MUTUAL INFORMATION AND SELF-SUPERVISED LEARNING TASKS

A core objective of machine learning is to learn effective data representations. Many methods attempt to To achieve this goal through maximizing mutual information (MI), e.g. InfoMax principle (Linsker, 1988) and information bottleneck principle (Tishby et al., 2000). Unfortunately, estimating MI is intractable in general (McAllester & Stratos, 2020). Therefore, many works resort to optimize the upper or lower bound of MI (Alemi et al., 2016; Poole et al., 2019; Ni et al., 2022)

In the field of self-supervised learning (SSL), there are two widely used methods for acquiring meaningful representations: contrastive methods and predictive (generative) methods. Recently, it has been discovered that these two methods are closely linked to the maximization of lower-bound mutual information (MI) targets. A summary of these relationships is presented below.

### B.2.1 CONTRASTIVE OBJECTIVE

Contrastive learning (CL) (Chen et al., 2020a) learn representations that are similar between positive pairs while distinct between negative pairs. From the perspective of mutual information maximization, CL actually maximizes the mutual information between the representations of positive pairs. The InfoNCE loss (Oord et al., 2018; Kong et al., 2019) is given by:

$$\mathcal{L}_{\text{InfoNCE}} = -E_{p(x,y)}\Big[\log \frac{f(x,y)}{\sum_{\tilde{y}\in\tilde{\mathcal{Y}}} f(x,\tilde{y})}\Big] \tag{23}$$

where $(x,y)$ is a positive pair, $\tilde{\mathcal{Y}}$ is the sample set containing the positive sample $y$ and $|\tilde{\mathcal{Y}}| - 1$ negative samples of $x$, $f(\cdot,\cdot)$ characterizes the similarity between the two input variables. (Oord et al., 2018) proved that minimizing the InfoNCE loss is maximizing a lower bound of the following mutual information:

$$I(X;Y) \geq \log|\tilde{\mathcal{Y}}| - \mathcal{L}_{InfoNCE}. \tag{24}$$

Denote $v_1$ and $v_2$ as two views of the input and $h_\theta$ is the representation function. Define $x = h_\theta(v_1)$ and $y = h_\theta(v_2)$ as representations of the two views and the similarity function $f(x,y) = \exp(x^\top y)$, contrastive learning is optimizing the following InfoNCE loss (Arora et al., 2019)

$$\mathcal{L}_{CL} = -E_{p(v_1,v_2^+,v_2^-)}\Big[\log \frac{exp(h_\theta(v_1)^T h_\theta(v_2^+))}{exp(h_\theta(v_1)^T h_\theta(v_2^+)) + \sum_{v_2^-} exp(h_\theta(v_1)^T h_\theta(v_2^-))}\Big], \tag{25}$$

where $v_2^+$ is the positive sample, $v_2^-$ is negative samples. Accordingly, minimizing the CL loss is maximizing the lower bound of $I(h_\theta(v_1), h_\theta(v_2))$ w.r.t. the representation function.

### B.2.2 PREDICTIVE OBJECTIVE (MASK-THEN-PREDICT)

The mask-then-predict task (Devlin et al., 2018) are revealed to maximize the mutual information between the representations of the context and the masked tokens (Kong et al., 2019). A lower bound of this MI can be derived in the form of a predictive loss:

$$\begin{aligned} I(X;Y) &= H(Y) - H(Y|X) \geq -H(Y|X) \\ &= E_{p(x,y)}\big[\log p(y|x)\big] \geq E_{p(x,y)}\big[\log q(y|x)\big]. \end{aligned} \tag{26}$$

The last inequation holds by applying the Jensen inequation $E_{p(x,y)}\big[\log \frac{q(y|x)}{p(y|x)}\big] \leq \log E_{p(x,y)}\big[\frac{q(y|x)}{p(y|x)}\big] = 0$.

Denote $x = h_\theta(c)$ and $y = h_\theta(m)$ as representations of the context $c$ and the masked token $m$ to be predicted. $q_\phi$ is the predictive model. This predictive objective $E_{p(c,m)}\big[\log q_\phi(h_\theta(m)|h_\theta(c))\big]$ corresponds to the training objective of a mask-then-predict task. Therefore, according to equation 26, mask-then-predict task maximizes the lower bound of the MI between representations of the context and the masked tokens, i.e.

$$I(h_\theta(C), h_\theta(M)) \geq E_{p(c,m)}\big[\log q_\phi(h_\theta(m)|h_\theta(c))\big]. \tag{27}$$

### B.2.3 GENERATIVE OBJECTIVE

(Liu et al., 2022b) conducts cross-modal pretraining by generating representations of one modality from the other. Utilizing equation 26 and the symmetry of mutual information, we can derive a lower bound of MI in the form of a mutual generative loss:

$$I(X;Y) \geq \frac{1}{2}E_{p(x,y)}\big[\log q(y|x) + \log q(x|y)\big]. \tag{28}$$

Denote $v_1$ and $v_2$ as two views of the input. $h_\theta$ is the representation function and $q_\phi$ is the predictive model. In equation 28, let $x = h_\theta(v_1)$ and $y = h_\theta(v_2)$, then we can derive that learning to generate the representation of one view from the other corresponds to maximize the lower bound of mutual information between the representations of the two views:

$$I(h_\theta(V_1), h_\theta(V_2)) \geq \frac{1}{2}E_{p(v_1,v_2)}\big[\log q_{\phi_1}(h_\theta(v_1)|h_\theta(v_2)) + \log q_{\phi_2}(h_\theta(v_2)|h_\theta(v_1))\big]. \tag{29}$$

### B.2.4 MODALITY BLENDING

We next present an theoretical understanding of multimodal blend-then-predict. For simplicity, we consider two relations, denoted as $\mathcal{R}_{\text{2D}} = (a_{ij})_{n \times n}$ and $\mathcal{R}_{\text{3D}} = (b_{ij})_{n \times n}$. Their elements are randomly partitioned into two parts by random partition variable $S$, represented as $\mathcal{R}_{\text{2D}} = [A_1, A_2], \mathcal{R}_{\text{3D}} = [B_1, B_2]$, such that $A_i$ shares identical elements indexes with $B_i$, $i \in \{1, 2\}$. The blended matrix is denoted as $\mathcal{R}_{\text{2D\&3D}} = [A_1, B_2]$. Our objective is to predict the two full modalities from the blended relations:

$$\max_{\theta, \phi_1, \phi_2} E_S E_{p(a_1, a_2, b_1, b_2)}[\log q_{\phi_1}(h_\theta(a_2)|h_\theta(a_1), h_\theta(b_2)) + \log q_{\phi_2}(h_\theta(b_1)|h_\theta(a_1), h_\theta(b_2))], \quad (30)$$

where $h_\theta$ is the representation extractor, $q_{\phi_1}$ and $q_{\phi_2}$ are predictive head that recovers $\mathcal{R}_{2D}$ and $\mathcal{R}_{3D}$. Utilizing the result from equation 27, the blend-then-predict objective aims to maximize the lower bound of mutual information presented below:

$$E_S I(h_\theta(A_2); h_\theta(A_1), h_\theta(B_2)) + I(h_\theta(B_1); h_\theta(A_1), h_\theta(B_2)). \quad (31)$$

From the mutual information decomposition in Proposition B.1, the objective in equation 31 can be divided into two parts.

$$E_S \Big\{ \frac{1}{2} \underbrace{[I(A_1; B_1) + I(A_2; B_2)}_{\text{contrastive and generative}} + \underbrace{I(A_1; B_1|B_2) + I(A_2; B_2|A_1)]}_{\text{conditional contrastive and generative}}$$

$$+ \frac{1}{2} \underbrace{[I(A_1; A_2) + I(B_1; B_2)}_{\text{mask-then-predict}} + \underbrace{I(A_1; A_2|B_2) + I(B_1; B_2|A_1)]}_{\text{multimodal mask-then-predict}} \Big\} \quad (32)$$

The first part of Equation 32 corresponds to existing (conditional) contrastive and generative methods, which aim to maximize the mutual information between two corresponding parts ($A_i$ with $B_i$, $i \in \{1, 2\}$) across two modalities . The second part represents the (multimodal) mask-then-predict objectives, focusing on maximizing the mutual information between the masked and the remaining parts within a single modality.

This decomposition demonstrates that our objective unifies contrastive, generative (inter-modality prediction), and mask-then-predict (intra-modality prediction) approaches within a single cohesive *blend-then-predict* framework, from the perspective of mutual information maximization. Moreover, this approach fosters enhanced cross-modal interaction by introducing an innovative multimodal mask-then-predict target.

## C   EXPERIMENTAL DETAILS

### C.1   DATASETS DETAILS

**MoleculeNet (Wu et al., 2017)**   11 datasets are used to evaluate model performance on 2D tasks:

- BBBP: The blood-brain barrier penetration dataset, aims at modeling and predicting the barrier permeability.
- Tox21: This dataset ("Toxicology in the 21st Century") contains qualitative toxicity measurements for 8014 compounds on 12 different targets, including nuclear receptors and stress response pathways.
- ToxCast: ToxCast is another data collection providing toxicology data for a large library of compounds based on in vitro high-throughput screening, including qualitative results of over 600 experiments on 8615 compounds.
- SIDER: The Side Effect Resource (SIDER) is a database of marketed drugs and adverse drug reactions (ADR), grouped into 27 system organ classes.
- ClinTox: The ClinTox dataset compares drugs approved by the FDA and drugs that have failed clinical trials for toxicity reasons. The dataset includes two classification tasks for 1491 drug compounds with known chemical structures: (1) clinical trial toxicity (or absence of toxicity) and (2) FDA approval status.

Table 8: Hyperparameters setup for pretraining.

| Hyperparameter | Value |
|---|---|
| Max learning rate | 1e-5 |
| Min learning rate | 0 |
| Learning rate schedule | cosine |
| Optimizer | Adam |
| Adam betas | (0.9, 0.999) |
| Batch size | 4096 |
| Training steps | 1,000,000 |
| Warmup steps | 100,000 |
| Weight Decay | 0.0 |
| num. of layers | 12 |
| num. of attention heads | 32 |
| embedding dim | 768 |
| num. of 3D Gaussian kernel | 128 |

Table 9: Search space for MoleculeNet tasks. Small datasets: BBBP, BACE, ClinTox, Tox21, Toxcast, SIDER, ESOL FreeSolv, Lipo. Large datasets: MUV.

| Hyperparameter | Small | Large | HIV |
|---|---|---|---|
| Learning rate | [1e-6, 1e-4] | [1e-6, 1e-4] | [1e6, 1e-4] |
| Batch size | {32,64,128,256} | {128,256} | {128,256} |
| Epochs | {40, 60, 80, 100} | {20, 40} | {2, 5, 10} |
| Weight Decay | [1e-7, 1e-3] | [1e-7, 1e-3] | [1e-7, 1e-3] |

- MUV: The Maximum Unbiased Validation (MUV) group is another benchmark dataset selected from PubChem BioAssay by applying a refined nearest neighbor analysis, containing 17 challenging tasks for around 90,000 compounds and is specifically designed for validation of virtual screening techniques.

- HIV: The HIV dataset was introduced by the Drug Therapeutics Program (DTP) AIDS Antiviral Screen, which tested the ability to inhibit HIV replication for over 40,000 compounds.

- BACE: The BACE dataset provides qualitative binding results for a set of inhibitors of human $\beta$-secretase 1. 1522 compounds with their 2D structures and binary labels are collected, built as a classification task.

- ESOL: ESOL is a small dataset consisting of water solubility data for 1128 compounds.

- FreeSolv: The Free Solvation Database provides experimental and calculated hydration free energy of small molecules in water.

- Lipo: Lipophilicity is an important feature of drug molecules that affects both membrane permeability and solubility. This dataset provides experimental results of octanol/water distribution coefficient (logD at pH 7.4) of 4200 compounds.

**QM9 (Ramakrishnan et al., 2014)** QM9 is a quantum chemistry benchmark consisting of 134k stable small organic molecules, corresponding to the subset of all 133,885 species out of the GDB-17 chemical universe of 166 billion organic molecules. The molecules in QM9 contains up to 9 heavy atoms. Each molecule is associated with 12 targets covering its geometric, energetic, electronic, and thermodynamic properties, which are calculated by density functional theory (DFT).

C.2 HYPERPARAMETERS

Hyperparameters for pretraining and finetuning on MoleculeNet and QM9 benchmarks are presented in Table 8, Table 9 and Table 10, repectively.

Table 10: Hyperparameters for QM9 finetuning.

| Hyperparameter | QM9 |
|---|---|
| Peak Learning rate | 1e-4 |
| End Learning rate | 1e-9 |
| Batch size | 128 |
| Warmup Steps | 60,000 |
| Max Steps | 600,000 |
| Weight Decay | 0.0 |

Table 11: Ablation studies on fintuning settings of 2D tasks.

| Finetuning Settings | BBBP ↑ | Tox21 ↑ | ToxCast ↑ | ClinTox ↑ | Bace ↑ | ESOL ↓ | FreeSolv ↓ | Lipo ↓ |
|---|---|---|---|---|---|---|---|---|
| 2D | **73.0** | **77.8** | 66.1 | 87.6 | 83.7 | **0.831** | 1.910 | 0.638 |
| 2D + 3D | 71.8 | 76.8 | **67.4** | **90.9** | **84.3** | 0.874 | **1.824** | **0.636** |

# D    ABLATION STUDIES

## D.1    2D TASKS WITH 3D INFORMATION

Since our model is pretrained to predict both 2D and 3D information, for 2D tasks, we consider utilizing the 3D information predicted by our model as supplementary information (2D + 3D in Table 11). We observe that both settings achieve comparable performance across various tasks. This may be due to the 2D and 3D spaces have been well aligned and 3D knowledge is implicit injected into the model, allowing it to achieve satisfactory results even with only 2D information provided.

