# OpenReview forum: "Multimodal Molecular Pretraining via Modality Blending"
_ICLR.cc/2024/Conference — ICLR 2024 poster_

### Official Review · Reviewer_3A6t · 2023-10-13

**Soundness:** 3 good
**Presentation:** 3 good
**Contribution:** 3 good
**Rating:** 6
**Confidence:** 4

**Summary:**

This paper seeks to understand the inherent connection between 2D and 3D representations, capturing the essential structural attributes of molecules through atomic-relation level multimodal pretraining techniques.
In this process, the authors initially combine atom relations from various modalities into a single cohesive matrix for combined encoding, and subsequently retrieve specific information for both 2D and 3D structures separately.

**Strengths:**

1. Learning the qualified representation of molecules is important for various downstream tasks.
2. To the best of my knowledge, this is the first work that aligns atom-level representation for 2D and 3D representations of molecules.

**Weaknesses:**

1. As highlighted by the authors, earlier studies typically aligned different modalities at a broader molecule level, potentially hindering the capture of detailed molecular structures. Initially, when considering atom-level depictions of a molecule, our focus might be on the atom-level rather than the atomic-relation level. For instance, we might design a model that determines 3D atom coordinates based on a 2D molecular graph. What's the rationale behind emphasizing atomic-relation level multi-modal pretraining? It would be beneficial to provide a thorough reasoning in the methodology section and draw empirical comparisons in the experiments.

2. The experimental findings appear to be underwhelming. MoleBlend's performance, as shown in Tables 1 and 2, doesn't seem to fare well against prior studies.

**Questions:**

Provided above.

---

> ### Author Response · Authors · 2023-11-20
> **Response to Reviewer 3A6t**
>
> We express our gratitude to the reviewer for providing valuable feedback. Below we address the comments in detail.
>
> **Q1: Why atomic-relation level rather than atomic-relation level.**
>
> We thank the reviewer for the question.
>
> In our analysis and intuitive insights, different modalities share the identical atoms (C, N, O, etc) and **diverse atomic relations**. Upon closer examination, the modalities mainly differ in how they depict **atom relations**, while the atoms are commonly shared. To this end, to align different modalities, we propose that the key to aligning different modalities lies in aligning the distinct atom relations that encapsulate richer information from various perspectives, rather than focusing solely on the identical atoms.
>
> **Predicting atom coordinates implicitly learns atom relations.** Regarding the coordinates mentioned, we posit that the individual atom coordinates themselves may not hold primary significance. Instead, the **relation/interplay between atoms (i.e., atom distance in 3D space, the structure)** imbues the molecule with diverse properties [1]. While predicting all coordinates collectively offers a degree of insight into structures and implicitly learns the atom relations, we contend that focusing on atom relations, specifically the 3D directed distances, **more directly and accurately convey the structural information**.
>
> At present, we recognize that the effectiveness of predicting atom coordinates lies in its implicit learning of structure and atom relations. The rationale behind the superiority of atom-level predictions over atom-relation level remains elusive. To this end, experiments exploring atom-level alignment falls beyond the scope of our current paper and can be left as future work.
>
> If any aspects require further clarification, please feel free to communicate them to us. We are eager to engage in additional discussions and appreciate your consideration. Thank you.
>
> [1] Structure-based drug design with geometric deep learning.
>
> **Q2: The experimental findings appear to be underwhelming.**
>
> Our method achieves state-of-the-art results across 16 out of 23 mainstream tasks evaluated. This is a substantial rather than marginal improvement. Notably, in Table 1, we achieve an average improvement of 1.09 point across all 8 MolNet tasks over the prominent SOTA MoleculeSDE [1] from ICML 2023. This improvement is obvious and far from being considered marginal.
>
> Besides, different from current 2D+3D baselines [1][2][3] that all rely on strong established methods like contrastive learning to enhance their performance, our method is a de novo endeavor -- the first attempt to align modalities at atom-relation level. We think these results are particularly encouraging, considering our distinctive perspective that offers a fresh outlook on the problem.
>
> [1] A group symmetric stochastic differential equation model for molecule multi-modal pretraining. ICML 2023
>
> [2] Pretraining molecular graph representation with 3d geometry. ICLR 2022
>
> [3] 3d infomax improves gnns for molecular property prediction. ICML 2022

---

> > ### Comment · Reviewer_3A6t · 2023-11-22
> >
> > We thank the authors for their detailed comments.
> >
> > Regarding Q1, I wonder if the authors can share their analysis and intuitions that the major difference exists in the depiction of the relationship. I disagree with the authors in the sense that the main difference exists in the relationship between atoms, while atoms are commonly shared. Can you clarify this part more specifically?
> >
> > In my opinion, I believe in a 2D graph, the atom is commonly represented by its atomic number and corresponding characteristics, while in a 3D structure, atom coordinates play an important role. I believe the atom itself is different.
> >
> > Moreover, I additionally found a relevant baseline that predicts atomic level representation with multiple modalities, 2D and 3D [1]. I think this work should also be included in related works and experiments to demonstrate the importance of learning the **relationship** between atoms.
> >
> > [1] Unified 2D and 3D Pre-Training of Molecular Representations, 2023 KDD.

---

> ### Author Response · Authors · 2023-11-22
> **Response to Reviewer 3A6t**
>
> We thank Reviewer 3A6t for the further comment and address the further concerns below.
>
> **Q1: Why the key of modality alignment exists in the atom relationships.**
>
> The reviewer's major concern lies in that atom coordinates play a key role. We argue that **when considering coordinates as attributes, what genuinely matters is the relative relation between the coordinates, i.e., the atom relation.**
>
>  **The coordinate themselves does not matter**, as supported by the fact that when we rotate a molecule, its properties remain constant and invariant while all coordinates changed. The true determinant of a molecule's property is **the invariant atom relations** during rotation.
>
> To this end, the employment of coordinates is typically coupled with equivariant constraints [1][2]. We posit that in essential, coordinate + equivariant constraints captialize on **atom relations**. Both the equivariant operation for processing coorinates (Figure 1 of [1], [2]) and the equivariant head used for predicting coordinates (Eq 6 of [3]) leverages **atom relations to guarantee equivariance** [1][2][3]. It is crucial to reiterate that coordinates must be coupled with equivariance for modeling molecules. A theoretical study [4] deduces that equivariance in essential can be expressed by relative scalars (e.g. relative distance), underscoring the significance of relative atom relations (or the equivalance of atom relations to equivariance and invariance).
>
> In summary, equivariance and invariance are indispensable for molecule properties, and atom relations are the key for ensuring equivariance and invariance. Notably, atom coordinates themselves lack inherent invariance and equivariance.
>
> We sincerely thank the reviewer for the question, and the relevant reference and rationale will be organized and included in our updated paper. We seek clarification on whether the reviewer agrees with this point.
>
> [1] SE(3)-Transformers: 3D Roto-Translation Equivariant Attention Networks. NIPS 2020
>
> [2] E(n) Equivariant Graph Neural Networks. ICML 2021
>
> [3] One Transformer can understand both 2d & 3d molecular data. ICLR 2023
>
> [4] Scalars are universal: Equivariant machine learning, structured like classical physics. NIPS 2021
>
> **Q2: Baseline [1]**
>
> In our experiments, we did not compare with [1], as we find problems existing in their evaluation procedure, and their MoleculeNet finetuning practice in the official repository differs from mainstream methods [2][3]. For example, they finetune and evaluate ClinTox on only one sub-task (https://github.com/teslacool/UnifiedMolPretrain/blob/7f299f65433f9f1b7ceef5a003315640ae0cec50/pretrain3d/data/DCGraphPropPredDataset/deepchem_dataloader.py\#L16), whereas mainstream evaluation trains on two sub-tasks simultaneously with a single head, which is much more difficult. We also struggled to evaluate it under the commonly adopted evaluation protocol as it did not provide a pretrained checkpoint. To this end, we opt not to include its numbers as our baseline. However, we have discussed their method in related work in our initial submission.
>
> The state-of-the-art work MoleculeSDE [2] from ICML 2023 also does not include its results as a baseline.
>
> [1] Unified 2D and 3D Pre-Training of Molecular Representations. KDD 2022
>
> [2] A group symmetric stochastic differential equation model for molecule multi-modal pretraining. ICML 2023
>
> [3] Pretraining molecular graph representation with 3d geometry. ICLR 2022

---

> > ### Comment · Reviewer_3A6t · 2023-11-22
> >
> > I appreciate the author for further clarifying the paper.
> >
> > Thanks to the author's clarification, it becomes more clear that the relationship (spd for 2D and distance for 3D) between the atoms is different between modalities.
> >
> > However, I still believe the baseline [1] model should be included in experiments to demonstrate the importance of learning from the relationship between the atoms.
> >
> > Therefore, I will keep my score.
> >
> > [1] Unified 2D and 3D Pre-Training of Molecular Representations. KDD 2022

---

> > > ### Comment · Reviewer_TmmD · 2023-11-22
> > > **A Quick Comment**
> > >
> > > Hi reviewer 3A6t,
> > >
> > > Thank you for pointing out this KDD paper. I read this, and I know that you are definitely not related.
> > >
> > > I just want to kindly point out that the 2D to 3D generative pretraining in this work (which is called DMCG) does not satisfy the SE(3)-equivariance property. This should be pointed out by the reviewers of KDD, but it seems they just missed this part.
> > >
> > > I am happy to discuss this in more detail.
> > >
> > > Regards

---

> > > ### Author Response · Authors · 2023-11-23
> > > **Follow-up. Thank you for the review again.**
> > >
> > > Dear Reviewer 3A6t,
> > >
> > > We sincerely extend our appreciation to you for dedicating valuable time to review our paper. **We have provided thorough clarifications to all questions, including 1) clarifying the motivation behind aligning atom relations, and 2) incorporating results and discussion of [1] into our manuscript.**
> > >
> > > Given the approaching deadline in several hours for discussions, we would like to confirm whether our responses have adequately addressed your questions. Should you have any additional questions or if further clarification is needed, please feel free to notify us. We are more than willing to promptly address any concerns you may have.
> > >
> > > Your time and consideration are greatly valued. Thank you for the engagement in the constructive dialogue again.
> > >
> > > Best regards,
> > >
> > > Authors

---

> ### Comment · Reviewer_3A6t · 2023-11-22
>
> Hello reviewer TmmD!
>
> Thank you for pointing out that the paper does not satisfy the SE(3)-equivariance property.
> However, I believe the inclusion of this baseline (or even a training approach as an ablation study) can definitely show the importance of learning from the relationship between atoms is importance, and improve the quality of the work.
>
> I'm happy to further listen to your opinion on this.

---

> > ### Comment · Reviewer_TmmD · 2023-11-22
> > **Reply**
> >
> > Hi reviewer 3A6t,
> >
> > Sure, I am happy to share more thoughts on this!
> >
> > So the way I viewed this is that starting from GraphMVP and 3D InfoMax, people have been working on 2D-3D pretraining, which somehow learns/gives the relationship between atom representations. Just to double-check, is the relationship you are referring to?
> >
> > If not, feel free to correct me :)
> >
> > If so, then the following works (like the KDD paper you shared and MoleculeSDE) are doing generative pretraining in the data space (2D & 3D graphs) instead of representation space. This gives a more direct way of showing the relationships between atoms. BTW. The MoleculeSDE papers are compared in this work.
> >
> > Feel free to share your thoughts!
> >
> > Regards

---

> ### Comment · Reviewer_3A6t · 2023-11-22
>
> Dear reviewer TmmD
>
> What I mean by relationship here is the relationship between atoms defined in this paper, i.e., distance, spd, edge type.
>
> As my understanding, GraphMVP and 3D Infomax are training the model by learning the representation of molecule by **molecule-level** alignment, while this paper aims to learn the representation by aligning **atom-level** representation.
>
> To the best of my knowledge, the KDD paper is the only paper with **atom-level** alignment, and this paper learns the representation by directly generating in data space without regarding **relationship**(i.e., distance, spd, edge type).
> Therefore, I believe the key difference between this paper and the KDD paper is regarding the relationship between the atoms, and the inclusion of this would improve the paper.
>
> I'm happy to be corrected if there is something wrong.
> Feel free to share your thoughts!
>
> **Also, for authors, if there is something wrong in my knowledge, please don't hesitate to correct it.**

---

> ### Author Response · Authors · 2023-11-22
> **Thank you for the prompt response.**
>
> We sincerely appreciate the prompt response from Reviewer 3A6t. We have incorporated the results from [1] on partial MolNet tasks in Table 1. However, it is important to note that these results are 1) incomplete MolNet results and 2) have been evaluated under a distinct protocol. Therefore, the comparability of these results to other baselines and our work may be limited.
>
> Concretely, one most obvious deviation is that the common protocal [2][3][4] of finetuning on MolNet is multitask training (as shown in Table 8 of GraphMVP [2] and Table 6 of Mole-BERT [4], for example, Tox21 and ClinTox are trained on 12 and 2 sub-tasks jointly). However, as per [1]'s official github repo, they focus only on one sub-task: ClinTox (https://github.com/teslacool/UnifiedMolPretrain/blob/7f299f65433f9f1b7ceef5a003315640ae0cec50/pretrain3d/data/DCGraphPropPredDataset/deepchem_dataloader.py#L16), Tox21 (https://github.com/teslacool/UnifiedMolPretrain/blob/7f299f65433f9f1b7ceef5a003315640ae0cec50/pretrain3d/data/DCGraphPropPredDataset/deepchem_dataloader.py#L34).
>
> We follow the [2][3][4]'s protocal for evaluation, which is also the mainstream protocal adopted by the community. To this end, our results are not comparable to [1]. However, we have discussed the methodology of this work in the related work section.
>
> We welcome the feedback on whether these adjustments adequately address the reviewer's concerns.
>
> Thank you.
>
> [1] Unified 2D and 3D Pre-Training of Molecular Representations. KDD 2022
>
> [2] Pretraining molecular graph representation with 3d geometry. ICLR 2022
>
> [3] A group symmetric stochastic differential equation model for molecule multi-modal pretraining. ICML 2023
>
> [4] Mole-BERT: Rethinking Pre-Training Graph Neural Networks for Molecules. ICLR 2023

---

> ### Author Response · Authors · 2023-11-22
> **Thanks for the discussion from Reviewer 3A6t and TmmD**
>
> We extend our sincere gratitude to the reviewers for your invaluable efforts and dedicated time invested in reviewing our paper.
>
> Let us organize the discussion thread to establish a clear logical progression.
>
> Our primary motivation is that aligning fine-grained atom-relations is crucial for aligning different modalities. This is well established in our introduction and the response to Reviewer TmmD and 3A6t. We thank Reviewers TmmD and 3A6t for the assistance in elucidating this point. Further, we have conducted theoretical analysis and experiments to validate the effectiveness of this fine-grained alignment, thereby supporting our motivation and claim.
>
> [1] explores the effectiveness of atom-level alignment, which is another form of fine-grained alignment of modalities. Reviewer 3A6t recommends that the comparison to [1] can enhance the quality of our paper. However, a notable challenge arises due to their adoption of a distinct evaluation protocol from mainstream studies and the absence of a provided pretrained checkpoint, making the evaluation of their method challenging. This lack of a standardized basis hinders a fair and quick comparison.
>
> It is imperative to underscore, however, that whether atom-level alignment is crucial extends beyond the current scope of our research and is less directly connected to our motivation. While we acknowledge the significance of this aspect, we believe that its thorough exploration can be left for future investigations.
>
> [1] Unified 2D and 3D Pre-Training of Molecular Representations. KDD 2022

---

### Official Review · Reviewer_w2xw · 2023-10-27

**Soundness:** 3 good
**Presentation:** 3 good
**Contribution:** 2 fair
**Rating:** 5
**Confidence:** 4

**Summary:**

The authors proposed to align molecule 2D and 3D modalities at the atomic-relation level and introduce MOLEBLEND. This multimodal molecular pretraining method explicitly utilizes the intrinsic correlations between 2D and 3D representations in pertaining. Extensive evaluation demonstrates that MOLEBLEND achieves state-of-the-art performance over diverse 2D and 3D tasks, verifying the effectiveness of relation-level alignment.

**Strengths:**

1. The paper is well-written and easy to understand.
2. The authors conducted extensive experiments on both 2D and 3D molecule tasks and showed their good performance.

**Weaknesses:**

1. The authors did not discuss the training time cost of different pretraining methods.
2. A series of pertaining baselines are missed in related works and comparisons. For example:

[1] Xu M, Wang H, Ni B, et al. Self-supervised graph-level representation learning with local and global structure. ICML 21.
[2] Zhang Z, Liu Q, Wang H, et al. Motif-based graph self-supervised learning for molecular property prediction. NeurIPS 21.
[3] Zaidi S, Schaarschmidt M, Martens J, et al. Pre-training via denoising for molecular property prediction. ICLR 23.

3. The theoretical analysis is good. However, could the authors provide more insights into why MOLBLEND overperforms existing methods theoretically?

4. How does the different choice of encodings for 2D/3D modalities influence the pretaining?

**Questions:**

Please see the weakness.

---

> ### Author Response · Authors · 2023-11-20
> **Response to Reviewer w2xw**
>
> We thank the reviewer for the helpful comments. We have uploaded a revision of our paper. Below we address the detailed questions.
>
> **Q1: Pretraining time cost**
>
> We thank the reviewer for the comment. The computation required are listed below. We can observe that GNN based pretrainig methods have relatively small compute cost. Transformer based pretraining methods cost a lot more computation.
>
> | Method      | Pretraining Time Cost     |
> | ----------- | ------------------------- |
> | AttrMask    | 9.2h on one V100 GPU      |
> | ContextPred | 23.3h on one V100 GPU     |
> | MolCLR      | 16.7h on one V100 GPU     |
> | InfoGraph   | 10h on one V100 GPU       |
> | GROVER      | 2.5 days on 250 V100 GPUs |
> | 3D Infomax  | Not reported              |
> | GraphMVP    | Not reported              |
> | MoleculeSDE | 50h on one V100 GPU       |
> | Ours        | 24 hours on 8 A100 GPUs   |
>
> **Q2: Missed baselines.**
>
> We thank the reviewer for the reference. [1][2] have been added into our paper for comparison.
>
> We do not compare with the number in [3] as they use a much stronger model backbone that excels at 3D property prediction, which even outperforms all Transformer-based pretraining method without pretraining.
>
> However, we implement the same denoising idea [3] on our backbone for comparison. In Table 3, where the first row is denoising (Noisy Node [4] is identical to [3] when applied to pretraining, as discussed in Section 2 and 3.2.2 of [3]), the second and the third row are our methods. The results show that our method achieve better results across a broad range of tasks.
>
> [1] Self-supervised graph-level representation learning with local and global structure. ICML 21.
>
> [2] Motif-based graph self-supervised learning for molecular property prediction. NeurIPS 21.
>
> [3] Pre-training via denoising for molecular property prediction. ICLR 23.
>
> [4] Simple GNN regularisation for 3d molecular property prediction and beyond. ICML 2022
>
> **Q3: More insights into why MOLBLEND overperforms existing methods theoretically.**
>
> Theoretically, the superiority of MoleBLEND lies in the more comprehensive and fine-grained optimization target for molecular modeling when compared with existing methods.
>
> - More comprehensive: as evidenced by Proposition 3.2, MoleBLEND can be viewed as a comprehensive target that encompasses single-modal masking, cross-modal contrastive and predictive methods, with their conditional versions. Therefore, MoleBLEND is able to learn cross-modal correspondences as well as single-modal context dependencies, which are both essential for multimodal molecular modeling. However, many existing methods focus solely on one aspect. For example, 3D InfoMax does not explicitly model the single-modal context dependencies.
> - More fine-grained: In the perspective of mutual information maximization, cross-modal contrastive and predictive methods aim at $\max I(A,B)$, where $A$ and $B$ are two modalities of the input. In contrast, MoleBLEND performs fine-grained modality alignment in Eq.15, i.e. $\max I(A_1,B_1)$, where $A_1$ and $B_1$ denote two modalities of the randomly partitioned input. This also makes the former objective a special case of ours. Fine-grained alignment encapsulates the cross-modal correspondence of molecular substructures, a critical characteristic of molecules that is rarely captured by previous methods.
>
> **Q4: How does the different choice of encodings for 2D/3D modalities influence the pretaining?**
>
> To the best of our knowledge, 2D shortest path, edge type, 3D euclidean distance are the most commonly used encodings for molecules. The only other encoding we know is molecule shape [1], but this encoding cannot be effectively represented in terms of atoms and their relationships, thus it is not compatible for our method. We acknowledge the potential existence of other encodings and are open to exploring their impact on pretraining. Should you have relevant references, we are willing to delve into further learning and are prepared to conduct swift experiments to assess their influence.
>
> [1] Learning Harmonic Molecular Representations on Riemannian Manifold. ICLR 2023

---

> ### Author Response · Authors · 2023-11-22
> **Anticipating Your Participation as Reviewer-Author Discussion Deadline Approaching**
>
> We express our gratitude to the reviewer for dedicating time to review our paper. **We have provided comprehensive clarifications to all the questions including 1) the pretraining time cost, 2) missed baselines, 3) theoretical insights of why MoleBLEND performs better, and 4) influences of different 2D/3D encodings.** As the discussion deadline looms within a day, we would like to inquire if our responses have adequately addressed your questions. We anticipate your participation in the Reviewer-Author discussion phase, as your insights are invaluable for refining and enhancing the quality of our paper. Should you have any additional queries or require further clarification, please do not hesitate to inform us. We are more than willing to address any concerns and ensure a comprehensive resolution. Thank you for your time and consideration.

---

### Official Review · Reviewer_TmmD · 2023-10-31

**Soundness:** 3 good
**Presentation:** 3 good
**Contribution:** 3 good
**Rating:** 8
**Confidence:** 3

**Summary:**

This paper proposes MolBlend, a method that explores the intrinsic alignment between 2D and 3D for molecule pretraining. MolBlend aims to conduct the 2D-3D pretraining based on the atom relations, which is finer-grained than previous works.

**Strengths:**

- The key idea is clear and straightforward: to use the attention module to help augment the 2D-3D atom-relation for molecule pretraining.
- The theoretical proof is interesting.

**Weaknesses:**

- The motivations are not clearly claimed or supported.
    - For instance, on Page 2, the authors say that they “observe that although appearing visually distinct … are intrinsically equivalent as they are essentially different manifestations of the same atoms and their relationships”. What does “equivalent” mean here? A lot of 2D-3D pretraining methods start by saying such two modalities are complementary to each other.
    - Additionally, on Page 2, what is the motivation to feed both modalities as one unified data structure to one single model in MoleBlend?

- Notations are misleading in Sec 3.1.
    - Why is $R_{spd}$ required? Because they can be derived from $R_{edge}$?
    - For Eq 1, the notation should be $R_{2D3D,S}$ (with S in the subscript).
    - Is $R_{2D3D}$ the masked version of $R_{spd}, R_{edge}, and R_{distance}$?

- Other minor comments:
    - The title can be further improved, especially that what modalities are considered is not explicit.
    - The distance modeling is invariant. A more advanced equivariant model is preferred here.
    - It would be better to explicitly add a column in the result tables on what backbone models are pretrained/comparing.
    - The citation of 3D InfoGraph is wrong. Please fix it.

I will consider raising the score once the authors answer/fix these comments.

**Questions:**

- I am confused about the Fig 1.b. Does this mean the input can be either 2D or 3D, and the output can be both 2D and 3D?

---

> ### Author Response · Authors · 2023-11-20
> **Response to Reviewer TmmD**
>
> We thank the reviewer for the positive feedback. Below we address your comments in detail.
>
> **Q1: What does 2D and 3D 'equivalent' mean? Motivation to feed both modalities as one unified data structure to one single model.**
>
> They are indeed complementary as existing 2D-3D pretraining methods claim. Furthermore, through our analysis and intuitive insights, we claim that they are essentially equivalent and just appear different.
>
> Different modalities share the same underlying structures, i.e., atoms and their relations. Different modalities share the identical atoms (C, N, O, etc). The 2D and 3D structures (edge type and 3D distance) are essentially different depictions of the **same** atom relations. To this end, we treat them as essentially equivalent. For example, a C-N bond in a 2D molecular graph and the corresponding 3D distance between C and N atoms essentially represent **the same thing**.
>
> Given they are essentially equivalant and complementary, these modalities should be amalgamated to form a more comprehensive information set. Consequently, we advocate for feeding these modalities as one unified data into one single model, to learn a comprehensive representation of molecules.
>
> If there are any aspects that need additional clarification, please let us know. We would be delighted to engage in further discussions. Thank you.
>
> **Q2: Notations Clarification**
>
> > Why is R_spd required? Because they can be derived from R_edge?
>
> We would like to clarify that R_spd and R_edge are two different types of information. R_spd is the shortest path between two atoms on a molecular graph, reflecting the topology information of the molecule. R_edge is the edge type (e.g. single bond), reflecting the chemistry property. Providing information from different perspectives can enhance model understanding of molecules.
>
> > For Eq 1, the notation should be (with S in the subscript).
>
> We thank the reviewer for the advice. We have updated this notation in our paper.
>
> > Is R_2d3d the masked version of R_spd, R_edge, R_distance?
>
> Yes, it is the blended and masked version of R_spd, R_edge, and R_distance.
>
> **Q3: Equivariant Head**
>
> We thank the reviewer for pointing this out and the advice. We conduct some preliminary attempts and transform our 3D prediction head to an equivariant head.
>
> Originally, to predict the 3D distance between atoms, we perform outer product operations between atoms representation, followed by flattening and projection operations to map them into the target space to predict the $N^2$ atom relations:
>
> $$o_{ij} = \text{G}(W_{l} X_i^{L+1}) \otimes \text{G}(W_{r} X_j^{L+1})^\top \in \mathbb{R}^{m\times m}$$
>
> $$z_{ij} = W_{\text{head}}\cdot \text{Flatten}(o_{ij}) \in \mathbb{R}^{c}$$
>
> Following [1], in order to achieve equivariant 3D relative position prediction, we decompose the flattened relational representation into three directions by multiplying the normalized relative position offset $\Delta_{i,j}=\frac{r_{i,j}}{\|\|r_{i,j}\|\|_2}$ and apply linear projection head to each component of the 3D relational representation in their respective direction:
>
> $$o_{ij} = \text{G}(W_{l} X_i^{L+1}) \otimes \text{G}(W_{r} X_j^{L+1})^\top \in \mathbb{R}^{m\times m}$$
>
> $$z_{ij}^{k} = W_{N}\cdot \Delta_{i,j}^{k} \text{Flatten}(o_{ij}), \quad  k=0,1,2$$
>
> where $\text{Flatten}(o_{ij})$ is the flattened relational representation, $\Delta_{i,j}^k$ is the k-th element of the directional vector $\frac{r_{i,j}}{\|\|r_{i,j}\|\|_2}$ between atom $i$ and atom $j$ and $W_N\in \mathbb{R}^{1\times d}$ is learnable weight matrices.
>
>
> We pretrain for 200K steps and compare it apple-to-apple with blending. The experiment results are shown below. Current equivariant head brings performance boost on 2 out of 4 tasks evaluated.
>
> | Method | BBBP | BACE | Tox21 | ToxCast |
> | --- | --- | --- | --- | --- |
> | Blending  | 71.68 | 83.41 | 76.58 | 65.46 |
> | Blending w/ Equivariant head | 72.60 | 81.55 | 76.41 | 66.03 |
>
> Please note that this is just a preliminary attempt, and further investigation can be left as future work.
>
> [1] One transformer can understand both 2d & 3d molecular data. ICLR 2023
>
> **Q4: Explanation of Fig 1 b**
>
> The output is still either 2D or 3D. The work of Fig 1 (b) is [1]. Their multimodal pretraining task involves taking a 2D input and predicting its corresponding 3D information, or taking a 3D input and predicting the associated 2D information, which corresponds to Fig 1 (b).
>
> [1] Unified 2d and 3d pre-training of molecular representations.
>
> **Q5: Other improvements**
>
> > It would be better to explicitly add a column in the result tables on what backbone models are pretrained/comparing.
>
> > The citation of 3D InfoGraph is wrong. Please fix it.
>
> We thank the reviewer for these suggestions. We have uploaded an updated version of our paper that addresses these questions.

---

> ### Comment · Reviewer_TmmD · 2023-11-20
> **Respone**
>
> Thank you. My questions have been addressed.

---

> > ### Author Response · Authors · 2023-11-20
> > **Thank you for the response.**
> >
> > We thank the reviewer very much for increasing the score. If there are any further inquiries or points for discussion, we remain open and available.

---

### Official Review · Reviewer_XaRC · 2023-10-31

**Soundness:** 3 good
**Presentation:** 3 good
**Contribution:** 2 fair
**Rating:** 5
**Confidence:** 2

**Summary:**

This paper introduces a molecular representation learning method that fusing the information from both 2D and 3D molecule structures. A unified relation matrix is constructed to describe the relationships between each pair of atoms, so that both 2D and 3D information can be injected into the matrix for fusion. For the 2D structure, based the bonds between atoms, shortest path and edge type information can be calculated for each entry of the relation matrix, and for the 3D structure, the entry can records the 3D Euclidean distance between atoms. The 2D and 3D information are blended in one relation matrix and a Transformer backbone is trained to recover the full information.

**Strengths:**

1. The idea of using a relation matrix to unify the 2D and 3D information for molecular representation learning is novel.
2. Theoretical analysis provide more insights to the proposed method.
3. The paper is well writen and structured and easy to follow.

**Weaknesses:**

1. The information gathered in the relation matrix is quite limited and much information in the original structure is lost, especially those in the 3D structure. The matrix construction is quite similar to the work of  "One transformer can understand both 2d & 3d molecular data" published in ICLR 2023.
2. Ablation studies on blending two masks should be provided.
3. Some details of the experimental setup is missing.

**Questions:**

1. Does the author run all baseline methods on the experimented splits or cite some results from other papers? For the results in Table 4, does the  single-modality mask-then-predict strategies use the same network as the proposed blending strategy?

2. In Table 4, I think the author compares blending three mask to using only one mask. What's the effect of blending two masks?

3. The conclusion in section D.1 is not well supported, since different 3D networks may perform differently and may have different speed of convergence. It's not enough to draw the conclusion by comparing the proposed method to only one 3D model.

4. Why run ablation studies on different datasets and different tasks?

---

> ### Author Response · Authors · 2023-11-20
> **Response to Reviewer XaRC (1/2)**
>
> We thank the reviewer for the helpful comments. Below we address each question in detail.
>
> **Q1: The information gathered in the relation matrix is limited and lost a lot.**
>
> We thank the reviewer for the question. We would like to clarify that no information is lost in our method in both the pretraining and finetuning stage.
>
> Please note that in modality-blending, we are performing self-/un-supervised pretraining. The learning task is to leverage the blended relation matrix to **predict non-blended elements**. Although some information in the original structure is not blended and lost, these elements will be **used as targets for prediction**. This supervision guarantees no information is lost in the process.
>
> During supervised finetuning and inference stage, we **utilize all available modalities without random blending, where no information in the original structure is lost**. The choice of modalities depends on the specific scenario at hand:
>
> - When a large amount of 2D molecular graphs are available while 3D conformations are too expensive to obtain, the model can be finetuned and perform inference with only 2D inputs.
> - When it comes to scenarios where 3D information is available, we propose to incorporate both 2D and 3D information as model input, as generating 2D molecular graphs from 3D conformations is free and can bring in useful information from 2D perspective.
>
> Detailed descriptions about how to incorporate each modality under different scenarios are presented in Section 3.3 of the paper. Our modality-blending pretraining technique enables the model to be versatile and adaptable to various downstream scenarios.
>
> If there are any aspects that need additional clarification, please let us know. We would be delighted to engage in further discussions. Thank you.
>
> **Q2: The matrix construction is similar to Transformer-M.**
>
> We thank the reviewer for the comment. It is imperative for us to elucidate that our primary emphasis lies in the development of an innovative "pretraining algorithm", and follow the model backbone of Transformer-M [1] and Graphformer [2]. In Section 3.2, we explicitly cite that we follow Transformer-M and Graphformer to choose 2D Shortest Path distance, 2D Edge Type, and 3D Euclidean distance as atom-relation encodings.
>
> It is crucial to underscore that our focus lies in improving the **incorporation** of the given 2D and 3D information, while the specific choice of 2D and 3D features is not our focus. Our method is generic and can be applied to any given 2D and 3D matrix, orthogonal to the specific choice.
>
> To draw an analogy, our work can be likened to the relationship of BERT to Transformer. In the realm of pretraining, our concentration is the formulation of an effective training algorithm. The design of the model backbone and data encoding falls outside the scope.
>
> [1] One transformer can understand both 2d & 3d molecular data. ICLR 2023
> [2] Do transformers really perform badly for graph representation? NIPS 2021
>
> **Q3: Ablations on blending two masks.**
>
> We thank the reviewer for the suggestion. The ablation results are presented in the table below, showing that blending 3 kinds of atom relations achieves best performance.
>
> | Method | BBBP | Bace | Tox21 | ToxCast |
> | --- | --- | --- | --- | --- |
> | SPD mask | 68.95  | 80.64  | 75.59  | 62.82 |
> | Edge mask  | 69.02  | 81.97  | 76.01  | 63.81 |
> | 3D mask   | 67.60  | 80.35  | 75.65  | 63.28 |
> | SPD_EDGE | 70.56 | 82.37 | 75.07 | 65.12 |
> | EDGE_3D | 71.12 | 81.43 | 76.07 | 65.10 |
> | SPD_3D | 71.23 | 79.1 | 75.90 | 65.04 |
> | Blending   | **71.68** | **83.41** | **76.58** | **65.46** |
>
> **Q4: details of the experimental setup.**
>
> We have made revisions to our paper (Section A.1). Below we address the detailed questions respectively.
>
> > Baseline results:
>
> - GraphMVP, MoleSDE, GraphCL, GraphMAE are from their own paper.
> - AttrMask, ContextPred, InfoGraph, MolCLR from MoleculeSDE
> - MoleBERT, 3D Infomax are from MoleBERT
> - The results of GROVER are from Uni-Mol
>
> All compared methods use the same dataset split.
>
> > Table 4, does the single-modality mask-then-predict strategies use the same network as the proposed blending strategy?
>
> Yes, the compared experiments use the same network. All settings are the same, isolating training objective as the varying factor for ablation.

---

> ### Author Response · Authors · 2023-11-20
> **Response to Reviewer XaRC (2/2)**
>
> **Q5: The conclusion in section D.1 is not well supported.**
>
> We thank the reviewer for the comment. We sincerely agree that more experiments are needed to support such conclusions. Due to this is not our primary focus and the convergence comparison with more 3D models needs further detailed investigation, we delete this experiment from our paper and leave it as future work.
>
> **Q6: Why run ablation studies on different datasets and different tasks?**
>
> We thank the reviewer for the question. For our final model, we evaluate it on a broad range of tasks (8 MolNet classification tasks, 3 MolNet regression tasks, and 12 QM9 tasks).
>
> For ablations, we assess partial tasks for two reasons: 1) the conclusion typically shares across different tasks, and can already be apparently established on conducted experiments; 2) conducting ablations on all datasets and tasks would be prohibitively costly.
>
> For ablation of Table 3, we evaluate full 8 molnet classification tasks plus two QM9 tasks. As for the ablations in Table 4 and 5, where the conclusions are clear and easily established due to a substantial performance margin, we opt to evaluate only on a subset of tasks from MolNet and QM9. These selected tasks align with our internal development process and serve as a validation of our methodology. In these instances, we believe it is unnecessary to conduct evaluations on all tasks, given the evident and easily comprehensible conclusions arising from the observed substantial margins.

---

> ### Author Response · Authors · 2023-11-22
> **Anticipating Your Participation as Reviewer-Author Discussion Deadline Approaching**
>
> We express our gratitude to the reviewer for dedicating time to review our paper. **We have provided comprehensive clarifications to all the questions, e.g. 1) the information gathered in the relation matrix is limited, 2) the similarity to Transformer-M.** As the discussion deadline looms within a day, we would like to inquire if our responses have adequately addressed your questions. We anticipate your participation in the Reviewer-Author discussion phase, as your insights are invaluable for refining and enhancing the quality of our paper. Should you have any additional queries or require further clarification, please do not hesitate to inform us. We are more than willing to address any concerns and ensure a comprehensive resolution. Thank you for your time and consideration.

---

### Public Comment · ~Tengwei_Song1 · 2024-06-20
**Requesting Code Access**

Dear Authors,

I am interested in the technical details of this work. Are your code and pre-trained weights available? If possible, please kindly provide public code access. Thank you very much!

---

### Meta-Review · Area_Chair_xK5g · 2023-12-05

**Metareview:**

The paper contributes to the quickly growing field of representation learning for molecules. The main challenge in this field is identifying novel and enriching learning signals. In contrast to NLP of vision, there are no natural self-supervised tasks that achieve very significant performance boosts. From this perspective, it is critical to clearly articulate the novel aspects and perform detailed ablations. Such information is necessary for the ICLR and broader community to be able to build on the work and work towards more powerful foundational models for molecules.

The paper main contribution lies in the development of a new pretraining algorithm. This makes the impact of the paper clearer, as it can be combined fruitfully with any other backbone. It is also timely because as noted earlier, the field is in a dire need of novel learning signals, provided they are complementary to already existing pretraining objectives.

Through an ablation, Authors show evidence that blending all sources of information is beneficial to performance.

Three reviewers who were negative about the paper did read the rebuttal. Two acknowledged that it did not address their most important concerns. One raised the score to 6. One of the key concerns is the lack of comparison to [1], which was not addressed (the argument was that results in [3] use a much stronger backbone).

All in all, it is my pleasure to recommend acceptance. The paper adds to the toolbox of pretraining objectives and presents reasonable evidence that it is a useful addition through its novelty and utilization of all modalities in a unified manner. The question that is hard to answer based on the provided evidence is whether this pretraining will contribute to the creation of foundational models in chemistry in the future. Nevertheless, it is a solid contribution that will be of interest to the ICLR community.

Please make sure that you make your code and pre-trained weights available, if that’s possible. It seems to me that currently there are relatively few state-of-the-art generic models for molecules that have pre-trained weights available.

References

[1] Pre-training via denoising for molecular property prediction. ICLR 23.

**Justification For Why Not Higher Score:**

The paper's pretraining objective is not novel enough nor motivated enough from first principles to remain relevant with high probability in the longer timeframe.

**Justification For Why Not Lower Score:**

See meta-review.

---

### Decision · Program_Chairs · 2024-01-16

Accept (poster)